# Volatility Spillovers between Stock Market and Hedge Funds: Evidence from Asia Pacific Region

Sameen Fatima [1], Christopher Gan [1] and Baiding Hu [2,*]

1 Department of Financial and Business Systems, Lincoln University, Christchurch 7647, New Zealand
2 Department of Global Value Chains and Trade, Lincoln University, Christchurch 7647, New Zealand
* Correspondence: baiding.hu@lincoln.ac.nz

**Abstract:** This paper investigates the nature of volatility spillovers between stock returns and hedge funds returns in twelve Asia Pacific countries in the 1997–2018 period. The sample period encompasses sub periods, 1997 Asia financial crisis, 2008 Global financial crisis and 2010 Eurozone crisis; these sub periods were characterised by financial upheavals. We apply the EGARCH methodology to model volatility and volatility spillovers in and between the markets. Our results show that the volatility of stock returns does not affect the volatility of hedge funds returns; however, there are inconsistent evidence of unidirectional volatility spillover from hedge funds to stock market returns.

**Keywords:** stock returns; hedge funds; integration; volatility spillovers; EGARCH modelling

## 1. Introduction

This paper investigates the inter-dependence, volatility spillovers, and the linkages among stock, hedge fund, and precious metal markets collectively, which have not been investigated simultaneously. Thirdly, the study sample includes 12 countries in the Asia-Pacific region. It examines the degree of change in the financial market linkages of hedge funds, stock markets, and precious metal markets. This differs from the literature where the magnitude and direction of co-movement specific to the countries under study has not been examined simultaneously.

An understanding of these volatility spillovers is likely to provide important information for more effective policy formulation on international financial markets' trading, as well as for fund/investment managers in terms of devising more effective strategies to hedge their portfolio and diversify. In this case for example investors should consider that they can diversify their portfolio considering hedge funds market in Asia-Pacific region, but they will also need to consider hedging strategies that protect their investment from shocks that could originate from international markets.

Beirne et al. (2010) and Cao and Jayasuriya (2012) measured the level of integration among emerging hedge fund markets by studying a sample of hedge funds based on two separate databases: the Centre of International Securities and derivative markets (CISDM) and Hedge Funds/Commodities Trading Advisors (CTA Database). The study includes only hedge funds that report their strategy as "emerging markets" to incorporate the impact of different investment strategies on the performance of hedge funds on 31 July 2007. This represents a total of 446 funds in the CISDM database. Beirne et al. (2010) concluded there are more local than global linkages in emerging markets and that, post 2008, the former have become more robust.

Though Cao and Jayasuriya (2012) discussed the integration that exists in relation to hedge funds, there is still a lack of research that explains the degree of interlinkage among specific countries in the Asia-Pacific region during periods of financial turmoil, like the Asian Financial Crisis, the Global Financial Crisis, and the 2010 Eurozone Crisis.

Guesmi and Nguyen (2011) studied the linkages of the MENA stock markets using a time-varying approach to the global capital market movement 1996 to 2008. The authors

applied the Bekaert and Harvey (1995) International Capital Asset Pricing Model (ICAPM) to investigate time-varying integration, local stock markets, and the degree of covariance among local and global stock markets. The ICAPM model allows local market returns to fluctuate over time according to the covariance between local and global market returns. In a perfectly integrated market, the risk of covariance relates to the price whereas the risk of variance is associated to a strictly segmented market. The ICAPM is thus useful for valuing international financial assets because it allows researchers to combine two major models, i.e., the complete integration and complete segmentation models.

Like the Guesmi et al. (2015) study, Cheng et al. (2010) used the Fama and French (1996) ICAMP model to identify whether there was evidence of static international CAPM efficiency in MENA markets and whether these financial markets were integrated with, or segmented from, global stock markets. The asset-pricing model assumes that all markets are highly integrated. Details can be found in studies conducted on the capital asset pricing model see Harvey (1991). Both Guesmi et al. (2015) and Cheng et al. (2010) concluded that most financial markets in the MENA region are strongly isolated from global financial markets.

Our study focusses on investigating the levels and magnitude of connections and provides a comprehensive, detailed summary of active linkages among Asian-Pacific countries' stock markets, hedge funds, and precious metal markets over 20 years. The study also explores the exact degree and changes, if any, in the long-term interdependence among these markets pre, during, and post the Asian Financial Crisis (1997), Global Financial Crisis (2008), and Eurozone Crisis (2010). This study also investigates whether there are any changes in the causal relationships among the 12 Asian Pacific markets after these crises. To the best of our knowledge, this is the first that compares the linkages among stock markets, hedge funds and precious metal markets in the Asia-Pacific region.

The results will educate investors about the importance of including these markets in their portfolios and especially in examining whether financial markets behave differently during crises and uncertainty in a background of globalisation. It will also help investors and portfolio managers identify trends in financial market fluctuations, even if the origin of a financial crisis is outside the Asia-Pacific region.

Finally, our empirical investigation and findings have important implications for policymakers and short and long term domestic and international investors and corporations. Our findings will enable them to anticipate future repercussions and the strength of linkages among these markets.

This study will help investment and portfolio managers to incorporate and leverage interlinked fluctuations among different financial markets, and, ultimately, improve their portfolio performance. In short, they will be able to design investment strategies tailored to any interlinked fluctuations in portfolio returns.

## 2. Literature Review

The existence of linkages among advanced financial markets has been previously well documented and studied. Researchers investigating dynamic market linkages, which provide evidence of causal relationships, have also found significant volatility spillovers and price volatility across advanced markets Bae et al. (2000); Hamao et al. (1990); Koutmos and Booth (1995); Theodossiou and Lee (1993).

In the last few decades, there has been growth and development in global financial markets characterised by increased capital movement and international trade across borders. These features have led to the integration and co-movement of individual financial markets. As a result, stock markets in one country can be affected by apparent fluctuations in the financial markets of another country, affecting the former's performance and trends. All stock market returns are not only influenced by their past performance, but also by global news from other international stock markets Lin et al. (1994).

Martin (2006) studied the performance of hedge funds to investigate the correlation between stock markets and hedge fund performance. The author used a modified Sharpe

ratio (MSR) to examine autocorrelation and skewness among hedge fund returns and to account for higher moments in the hedge fund returns distribution. Martin's results suggest that hedge fund returns have a higher MSR than bond and stock markets. However, general hedge funds show low correlations with other asset classes. However, when the hedge fund returns are corrected for bias and fat tails, autocorrelation still remains for a few hedge funds. Thus, market neutral stocks remain an attractive investment with higher returns than stock market indices. Studies by Stulz (2007) and Balakrishna (2012) support the above claim.

Like claimed, Balakrishna (2012), and Stulz (2007) suggested that from 1994–2000, hedge funds underperformed in the stock market (S&P 500) However, after 2000, hedge funds began to consistently outperform stock markets. Hedge funds also exhibited lower levels of standard deviation Lhabitant (2011); Martin (2006). A recent study by Advisors (2015) using the Credit Suisse Dow Jones database, suggested that the DJCS aggregated hedge funds index outperformed the stock market (MSCI World Index) during all financial crises that occurred from 1994–2009. Advisors (2015) study covered until 2010 when the financial crisis was still ongoing.

This paper examines the performance and linkages of hedge funds with the stock markets until 2018. This period covers the 1997 Asia Financial Crisis, after the 2008 Global Financial Crisis and the 2010 Eurozone crisis.

The literature also highlights the risk-return profile of portfolios, including Agarwal and Naik (2000) and Amin and Kat (2003) who claim that there is a correlation between hedge funds and other asset classes. As Fung and Hsieh (1997) have suggested, exposure to hedge funds in an investment portfolio improves the risk-return profile of the investment and provides better asset allocation and portfolio diversification. We test this to see if we reach the same conclusion.

Guesmi et al. (2015) analysed the correlation between equities, bonds, and hedge funds. The study period covered the Asian Financial Crisis (1997), the (2000) Technology Bubble Crisis and the (2008) Global Financial Crisis. They found that, in general, some hedge funds in their study outperformed the passive benchmarks, making them attractive for investors looking to diversify their portfolios. High net worth individuals and institutional investors have invested significant amounts of money in hedge funds, seeking benefits associated with diversification and the high returns promised by hedge fund managers Fung et al. (2008).

During the 2007 US subprime loan crisis, the hedge fund market was unable to generate the same level of positive returns independent of market conditions. Therefore, in contrast to other similar researchers, Fung and Hsieh (1997) and Guesmi et al. (2015) found that hedge fund markets have a greater correlation with other asset classes, such as stock and bonds, than previously thought, especially during distressed financial market conditions. Their results showed that the 2008 Global Financial Crisis had the greatest negative impact on hedge funds.

There is also extensive literature on the performance of precious metals and the linkages between stock markets and precious metals during periods of financial turmoil. Several studies highlight the safe haven properties of precious metals, particularly gold, as a strong, short-run hedge against financial market distress Baur and McDermott (2010); Lucey and Li (2015).

### 3. Data and Models

Our analysis focused on 12 Asia-Pacific countries' (Australia, New Zealand, Japan, Singapore, India, South Korea, China, Hong Kong, Thailand, Indonesia, Malaysia, and Taiwan) stock markets to examine the relationship between stock price movements and market distress. The data was collected from the Bloomberg database, consist of weekly closing price values of the 12 Asia Pacific countries' indices (ASX Index, the NZX Index, the JPY Index, the STI Index, the NSE500 Index, the KOSPI Index, the SHCOMP Index, the HSI Index, the SET Index, the JCI Index, the FBMKLCI Index, and the TWSE Index). The

total sample comprises 13,150 weekly observations where 9 out of 12 Asia Pacific countries' indices under study covers the duration of 1 January 1997 to 31 December 2018. The rest of the three countries' indices such as ASX Index, NZX Index and STI Index cover the weekly available closing price values for the duration of 1998–2018, 2011–2018, and 1999–2018, respectively. The period includes three major financial crises: the 1997 Asia Financial Crisis; the 2008 Global Financial Crisis; and the 2010 Eurozone Crisis.

We have monthly hedge fund returns data obtained from Eurekahedge. Eurekahedge firm is a major hedge funds database provider and is used in multiple studies as a reliable data source for, e.g., Mableannchang (2020); Jon (2021); Lan et al. (2019); Wang (2016). To ensure consistency with the other variables, we used a linear interpolation in Eviews to impute the weekly observations. We did this to fill in gaps in hedge fund observations between monthly and weekly data points. As hedge funds report their data on a monthly and quarterly basis, we use the available data and convert them into weekly frequency. The reason for the high frequency of data is because it allows us to identify performance volatility without averaging out the price volatility because of lower data frequency points. This method of data interpolation has been widely used in finance studies to obtain data points for missing datasets, e.g., Blanco et al. (2005); Zhu (2006); Norden and Weber (2009).

To analyse the hedge funds in the Asia-Pacific region, we used only funds that were created and are currently operating in the Asia-Pacific region. This gave a total of 28,700 observations that we used to investigate the linkages in return volatility between hedge funds and stock markets. Since hedge fund returns are uncorrelated and also affected by the number of risk factors, we filtered the raw hedge funds data by first filtering the (AR1) models with augmented factors as suggested by Agarwal and Naik (2004); Fama and French (1993); Fung and Hsieh (2019). After using this filtration method, we then applied the residuals filtered returns) to reduce the possibility of autocorrelation in the return series.

As an initial step we provide descriptive statistics for stock returns and hedge fund returns, in order to summarise the statistical characteristics of our sample see Tables A1–A3 in the (Appendix A). We then proceed and perform a stationarity test on each of the relevant variables that are included in our analysis to ensure that the results from the analysis are not spurious. We apply the Dickey Fuller (DF) test or Augmented Dickey-Fuller test (ADF) procedure if serial correlation is present. We also apply the Lagrange Multiplier (LM) test, to ensure that a sufficient number of lags have been added in the ADF test to ensure that there is no serial correlation present and the results of the ADF test are valid. The LM test is applied given that it is valid in the presence of lagged dependent variables as well as having the advantage of testing for first and higher orders of serial correlation. We estimate a Vector Autoregression (VAR) model in order to select the number of lags that would be appropriate to apply to our variables. We estimate the lag selection tests up 20 lags. In terms of choosing between the various lag length selection criteria, we follow Johansen et al. (2000) who suggest that when different information criteria suggest different lag lengths, it is common practice to prefer Hannan–Quinn HQ) criteria. Again, we ensure that the lag length selected for the VAR model is free from serial correlation after performing by applying the LM test to test for serial correlation up to the number of lags in the VAR model. We then proceed with our volatility analysis and apply a bivariate extension of the EGARCH model in order to examine whether the volatility of stock returns affects and is affected by the volatility of hedge fund returns within each economy. The EGARCH specification Nelson (1991) is used in order to test whether the volatility spillover effects are asymmetric. For example, an asymmetric spillover from stock returns to hedge fund returns would suggest that the effect of "bad" stock market news on the hedge fund returns is greater than the effect of "good" news. A coefficient regarding volatility persistence implies any deviation in the market returns from its expected return cause the variance in returns to be larger than expected. This implies that the amplitude in the returns fluctuations represents the amount of variations of the returns in short term; this holds a significant consideration for risk management of the investment portfolio.

The VAR model below is assumed to capture the dynamics between the returns of the stock market and the hedge fund:

$$S_t = \alpha_{s,0} + \sum_{i=1} \alpha_{s,i}\, S_{t-i} + \sum_{i=1} \alpha_{H,i}\, H_{t-i} + e_{S,t} \tag{1}$$

$$H_t = \alpha_{H,0} + \sum_{i=1} \alpha_{H,i}\, P_{t-i} + \sum_{i=1} \alpha_{S,i}\, S_{t-i} + e_{H,t} \tag{2}$$

where: $S_t$ is stock returns and $H_t$ is hedge fund returns, and the lag lengths are determined by information criteria. The above notation will be applied to the rest of the equations throughout the methodology in this paper, where the error term is represented by:

$$e_{S,t} \mid \Omega_{S,t-1} \approx N\,(0,\, \sigma^2_{S,t}) \tag{3}$$

$$e_{H,t} \mid \Omega_{H,t-1} \approx N\,(0,\, \sigma^2_{H,t}) \tag{4}$$

The conditional variance of stock and hedge funds return is modelled by a EGARCH (1, 1) model as follows:

$$\mathrm{Log}\,\sigma^2_{S,t} = \exp\{c_{s,0} + b_S \log(\sigma^2_{S,t-1}) + \delta_{S,S}(\mid z_{S,t-1}\mid - E\mid z_{S,t-1}\mid + \theta_{S,S}\, z_{S,t-1}) + \delta_{S,H}(\mid z_{H,t-1}\mid - E\mid z_{H,t-1}\mid + \theta_{S,H}\, z_{H,t-1})\} \tag{5}$$

$$\mathrm{Log}\,\sigma^2_{H,t} = \exp\{c_{H,0} + b_H \log(\sigma^2_{H,t-1}) + \delta_{H,H}(\mid z_{H,t-1}\mid - E\mid z_{H,t-1}\mid + \theta_{H,H}\, z_{H,t-1}) + \delta_{H,S}[(\mid z_{S,t-1}\mid - E\mid z_{S,t-1}\mid + \theta_{H,S}\, z_{S,t-1})]\} \tag{6}$$

$$\text{where: } \sigma_{S,H,T} = \rho\,\sigma_{S,t}\,\sigma_{H,t} \tag{7}$$

We summarize each of the relevant terms in Equations (1)–(7) in Table 1.

**Table 1.** Description of Model Parameters.

| Measures | Stock Market | Hedge Fund |
|---|:---:|:---:|
| Stochastic error term | $e_{S,t}$ | $e_{H,t}$ |
| Information set at time t − 1 | $\Omega_{S,t-1}$ | $\Omega_{H,t-1}$ |
| Conditional time varying variances | $\sigma^2_{S,t}$ | $\sigma^2_{H,t}$ |
| Persistence of volatility | $b_S$ | $b_H$ |
| Standardised residuals assumed to be normally distributed with 0 mean and variances of $\sigma^2_{S,t}$, $\sigma^2_{H,t}$ | $Z_{S,t} = e_{S,t}/\sigma^2_{S,t}$<br>$e_{S,t}/\Omega_{t-1} \approx N(0, \sigma^2_{S,t})$ | $Z_{H,t} = e_{H,t}/\sigma^2_{H,t}$<br>$e_{H,t}/\Omega_{t-1} \approx N(0, \sigma^2_{H,t})$ |
| ARCH effect where the parameters $\theta_{S,S}$, $\theta_{H,H}$ allow the effect to be asymmetric | $[\mid z_{S,t}\mid - E\mid z_{S,t}\mid + \theta_{S,H}\, z_{S,t}]$ | $[\mid z_{H,t}\mid - E\mid z_{H,t}\mid + \theta_{H,S}\, z_{H,t}]$ |
| Volatility spillovers | $\delta_{S,H}[(\mid z_{H,t-1}\mid - E\mid z_{H,t-1}\mid + \theta_{S,H}\, z_{H,t-1})]$ | $\delta_{H,S}[(\mid z_{S,t-1}\mid - E\mid z_{S,t-1}\mid + \theta_{H,S}\, z_{S,t-1})]$ |
| Measure of spillovers | $\delta_{S,H}$ | $\delta_{H,S}$ |
| Asymmetry of spillovers | $\theta_{S,H}$ | $\theta_{H,S}$ |
| Correlation of coefficient for standardised residuals | $\rho$ | |

The number of lags for the conditional mean Equations are determined using the Hannan–Quinn (HQ) criterion which is preferable to the more commonly used Akaike's Information Criteria (AIC), as the latter tends to overparameterize the models. Next, we apply the likelihood ratio (LR) test to determine the lag truncation length, p. We perform separate LR test on the stock returns and hedge fund returns conditional variance Equations to determine the optimal lag length for the EGARCH specification of each equation.

## 4. Empirical Results

### 4.1. Descriptive Statistics

This section discusses the descriptive statistics of stock and hedge fund returns that constitute the study sample countries. For the entire study time-series, the sample means of stock returns for all 12 Asia-Pacific stock indexes are positive.[1] The highest mean value for the stock market indices was for Hong Kong, during the 1997 Asian Financial Crisis,

followed by Taiwan, Malaysia, and China, at that same time. During the 2008 Global Financial Crisis, Hong Kong exhibited the highest mean return, followed by Singapore, China, Malaysia, and Taiwan. Similar results were seen for the 2010–2018 Eurozone Crisis. This implies a strong price volatility effect among these countries specific to the financial crisis period and interpreted as the sign of instability. The 25 hedge fund market returns exhibit positive means (Table A2)[2]; the mean values were used to calculate the central tendency of the hedge fund returns during the study period. The results show IIF, IGF, VPC-A, CFB-T, AACF, VPC-B, IVI, and HFNV with highest mean values, suggesting price volatility during the sample period.

### 4.2. Standardised Residuals

The skewness and kurtosis results suggest that stock returns are platykurtic in relation to a normal distribution where all stock return distributions are negatively skewed. This means that investors can experience fewer fluctuations resulting in greater potential for less extreme returns at both the upper and lower end. These findings are similar to previous studies' results, e.g., Caporale et al. (2002). The JB test results were very high, indicating rejection of the null hypothesis of normally distributed stock returns for the study period.

The hedge fund returns had positive means. The skewness and kurtosis tests revealed that the returns are leptokurtic in relation to a normal distribution where all the stock return distributions show positive kurtosis (kurtosis > 3) see Appendix A (Table A2). This means that investors can experience broader fluctuations resulting in greater potential for extremely low or extremely high returns. Again, we found a large JB value indicating hedge fund returns are not normally distributed.

Next, we discuss the results obtained from the empirical models. The ADF Test results see Appendix A (Table A4)[3] suggest that we reject the null hypothesis of the unit root in levels; all series are (I0). Given the variables are interlinked at the same level, we can conclude that there are some linkages. These results suggest a long-term relationship between stock market prices and hedge fund prices. Next, we ran the likelihood ratio (LR) to obtain the lag length (p) for the conditional mean equations in the bivariate EGARCH model[4].

To check the validity of the assumption of constant correlation adopted in the estimation of the models, the LB statistics of the standardised residuals from the stock market and hedge funds' return equation are calculated and these statistics indicated that the assumption of constant correlation over time can be accepted in almost all cases. These exceptions are normally corrected after increasing or decreasing the number of lags in the test see Appendix A.

### 4.3. Volatility Persistence

With regard to the volatility persistence term coefficient, the results in (Tables 2–13) indicate that, with the exception of South Korea the (KOSPI Index), there is significant persistence in stock market returns volatility for all 12 Asia-Pacific sample countries during the study period. In terms of hedge funds' returns, the results show that the coefficients are all significant for volatility persistence. A necessary condition for volatility persistence is that the value of the estimated coefficient needs to be less than one Wu (2005). Our results satisfy this condition.

The significant results of volatility persistence have higher repercussions in those stock markets where the level of integration is higher, than in those stock markets where a lack of linkages exists.

Table 2 shows the results of Australian stock market's impact on the hedge funds market and vice versa. For volatility persistence, coefficient (b) is significant for the Australian stock market. The volatility persistence for hedge funds is also significant. In addition, the values of (bs) and ($b_H$) are less than one, a condition necessary to have stable volatility Wu (2005).

**Table 2.** Volatility spillover between the Australian Stock Market (ASX) and Hedge Fund Returns.

| Australia | AGF | AACF | AAGF | BDP | BAF |
|---|---|---|---|---|---|
| Volatility persistence stock returns $b_S$ | 0.996 | 0.996 | 0.996 | 0.997 | 0.995 |
| | 0.000 * | 0.000 * | 0.000 * | 0.000 * | 0.000 * |
| Spillover from stock returns to HF price $\delta_{S,H}$ | 0.117 | 0.118 | 0.120 | 0.125 | 0.130 |
| | 0.000 * | 0.000 * | 0.000 * | 0.000 * | 0.000 * |
| Asymmetric spillover effect from stock returns to HF price $\theta_{S,H}$ | 0.050 | 0.050 | 0.041 | 0.040 | 0.039 |
| | 0.001 * | 0.001 * | 0.002 * | 0.003 * | 0.004 * |
| Volatility persistence HF price $b_H$ | 0.465 | 0.596 | 0.594 | 0.419 | 0.446 |
| | 0.000 * | 0.000 * | 0.000 * | 0.000 * | 0.000 * |
| Spillover from HF price to stock returns $\delta_{H,S}$ | 1.649 | 1.667 | 1.333 | 1.382 | 1.365 |
| | 0.000 * | 0.000 * | 0.000 * | 0.000 * | 0.000 * |
| Asymmetric spillover effect from HF price to stock returns $\theta_{H,S}$ | 0.008 | −0.031 | −0.008 | 0.051 | 0.003 |
| | 0.924 | 0.704 | 0.890 | 0.501 | 0.967 |
| Correlation coefficient $\rho$ | −0.180 | −0.098 | −0.153 | −0.052 | −0.072 |
| **Australia** | **CFB-FE** | **CFB-T** | **CFB-HK** | **HFNV** | **HKP** |
| Volatility persistence stock returns $b_S$ | 0.996 | 0.995 | 0.996 | 0.996 | 0.996 |
| | 0.000 * | 0.000 * | 0.000 * | 0.000 * | 0.000 * |
| Spillover from stock returns to HF price $\delta_{S,H}$ | 0.130 | 0.120 | 0.127 | 0.142 | 0.120 |
| | 0.000 * | 0.000 * | 0.000 * | 0.000 * | 0.000 * |
| Asymmetric spillover effect from stock returns to HF price $\theta_{S,H}$ | 0.030 | 0.042 | 0.033 | 0.040 | 0.044 |
| | 0.035 ** | 0.008 * | 0.020 ** | 0.012 ** | 0.001 * |
| Volatility persistence HF price $b_H$ | 0.488 | 0.532 | 0.608 | 0.553 | 0.523 |
| | 0.000 * | 0.000 * | 0.000 * | 0.000 * | 0.000 * |
| Spillover from HF price to stock returns $\delta_{H,S}$ | 1.314 | 1.577 | 1.709 | 1.612 | 1.502 |
| | 0.000 * | 0.000 * | 0.000 * | 0.000 * | 0.000 * |
| Asymmetric spillover effect from HF price to stock returns $\theta_{H,S}$ | 0.030 | 0.008 | −0.047 | −0.059 | 0.017 |
| | 0.644 | 0.908 | 0.418 | 0.393 | 0.839 |
| Correlation coefficient $\rho$ | −0.206 | −0.141 | −0.207 | −0.009 | −0.109 |
| **Australia** | **IIF** | **IVI** | **IGF** | **JKAI** | **LIM** |
| Volatility persistence stock returns $b_S$ | 0.997 | 0.996 | 0.996 | 0.996 | 0.996 |
| | 0.000 * | 0.000 * | 0.000 * | 0.000 * | 0.000 * |
| Spillover from stock returns to HF price $\delta_{S,H}$ | 0.131 | 0.139 | 0.138 | 0.118 | 0.138 |
| | 0.000 * | 0.000 * | 0.000 * | 0.000 * | 0.000 * |
| Asymmetric spillover effect from stock returns to HF price $\theta_{S,H}$ | 0.041 | 0.028 | 0.031 | 0.055 | 0.032 |
| | 0.009 * | 0.053 ** | 0.038 ** | 0.000 * | 0.022 ** |
| Volatility persistence HF price $b_H$ | 0.490 | 0.553 | 0.597 | 0.480 | 0.514 |
| | 0.000 * | 0.000 * | 0.000 * | 0.000 * | 0.000 * |
| Spillover from HF price to stock returns $\delta_{H,S}$ | 1.508 | 1.549 | 1.733 | 1.421 | 1.284 |
| | 0.000 * | 0.000 * | 0.000 * | 0.000 * | 0.000 * |
| Asymmetric spillover effect from HF price to stock returns $\theta_{H,S}$ | 0.033 | 0.026 | 0.026 | 0.008 | −0.012 |
| | 0.650 | 0.681 | 0.754 | 0.925 | 0.887 |
| Correlation coefficient $\rho$ | −0.086 | −0.075 | −0.147 | −0.054 | −0.195 |

**Table 2.** *Cont.*

| Australia | MLM | PPL | PIF | PCF | ISF |
|---|---|---|---|---|---|
| Volatility persistence stock returns $b_S$ | 0.996 | 0.997 | 0.997 | 0.997 | 0.996 |
| | 0.000 * | 0.000 * | 0.000 * | 0.000 * | 0.000 * |
| Spillover from stock returns to HF price $\delta_{S,H}$ | 0.126 | 0.109 | 0.119 | 0.118 | 0.115 |
| | 0.000 * | 0.000 * | 0.000 * | 0.000 * | 0.000 * |
| Asymmetric spillover effect from stock returns to HF price $\theta_{S,H}$ | 0.036 | 0.059 | 0.041 | 0.054 | 0.054 |
| | 0.004 * | 0.000 * | 0.001 * | 0.000 * | 0.000 * |
| Volatility persistence HF price $b_H$ | 0.403 | 0.443 | 0.436 | 0.544 | 0.494 |
| | 0.000 * | 0.000 * | 0.000 * | 0.000 * | 0.000 * |
| Spillover from HF price to stock returns $\delta_{H,S}$ | 1.317 | 1.488 | 1.547 | 1.761 | 1.390 |
| | 0.000 * | 0.000 * | 0.000 * | 0.000 * | 0.000 * |
| Asymmetric spillover effect from HF price to stock returns $\theta_{H,S}$ | −0.027 | −0.037 | 0.028 | 0.010 | −0.001 |
| | 0.737 | 0.659 | 0.757 | 0.915 | 0.993 |
| Correlation coefficient $\rho$ | −0.094 | −0.188 | −0.150 | −0.117 | −0.051 |
| **Australia** | **SJO** | **SRG** | **VPC-A** | **VPC-B** | **VEI** |
| Volatility persistence stock returns $b_S$ | 0.997 | 0.996 | 0.996 | 0.996 | 0.996 |
| | 0.000 * | 0.000 * | 0.000 * | 0.000 * | 0.000 * |
| Spillover from stock returns to HF price $\delta_{S,H}$ | 0.117 | 0.128 | 0.115 | 0.115 | 0.119 |
| | 0.000 * | 0.000 * | 0.000 * | 0.000 * | 0.000 * |
| Asymmetric spillover effect from stock returns to HF price $\theta_{S,H}$ | 0.042 | 0.041 | 0.046 | 0.047 | 0.039 |
| | 0.001 * | 0.003 * | 0.001 * | 0.001 * | 0.004 * |
| Volatility persistence HF price $b_H$ | 0.492 | 0.483 | 0.495 | 0.488 | 0.778 |
| | 0.000 * | 0.000 * | 0.000 * | 0.000 * | 0.000 * |
| Spillover from HF price to stock returns $\delta_{H,S}$ | 1.547 | 1.366 | 1.510 | 1.485 | 0.926 |
| | 0.000 * | 0.000 * | 0.000 * | 0.000 * | 0.000 * |
| Asymmetric spillover effect from HF price to stock returns $\theta_{H,S}$ | −0.064 | −0.021 | −0.040 | −0.041 | −0.010 |
| | 0.355 | 0.739 | 0.548 | 0.551 | 0.865 |
| Correlation coefficient $\rho$ | −0.066 | −0.074 | −0.153 | −0.154 | −0.003 |

* 1% significance level. ** 5% significance level.

The volatility spillover effect results shows that the coefficient ($\delta$) is significant, which suggests bidirectional volatility spillover exists between the Australian stock market and the hedge funds market. For the asymmetric spillover response, coefficient ($\theta$) is significant for the Australian stock market on the hedge funds market, but the reciprocal is insignificant. This implies that negative shocks from Australian stock market generate greater volatility in hedge fund markets than positive shocks of a similar magnitude.

Table 3 shows the results of the New Zealand stock market's impact on the hedge funds market and vice versa. For volatility persistence, the coefficient (b) is significant for the New Zealand stock market. The volatility persistence for hedge funds is also significant like for the Australian stock market. In addition, the values of ($b_S$) and ($b_H$) are less than one, a condition necessary to have stable volatility Wu (2005). As for volatility spillover, we find that the coefficient ($\delta$) is significant, which suggests bidirectional volatility spillover exists between the New Zealand stock market and the hedge fund market. For the asymmetric spillover response, the coefficient ($\theta$) is also significant for the New Zealand stock market and the hedge fund market, but the reciprocal is insignificant. This implies that negative shocks from the New Zealand stock market generate greater volatility in the hedge fund market than positive shocks of a similar magnitude.

**Table 3.** The Volatility spillover between the New Zealand Stock Market (NZX) and Hedge Fund Returns.

| New Zealand | AGF | AACF | AAGF | BDP | BAF |
|---|---|---|---|---|---|
| Volatility persistence stock returns $b_S$ | 0.842 | 0.842 | 0.817 | 0.842 | 0.841 |
| | 0.000 * | 0.000 * | 0.000 * | 0.000 * | 0.000 * |
| Spillover from stock returns to HF price $\delta_{S,H}$ | 0.099 | 0.099 | 0.098 | 0.105 | 0.110 |
| | 0.000 * | 0.000 * | 0.000 * | 0.000 * | 0.000 * |
| Asymmetric spillover effect from stock returns to HF price $\theta_{S,H}$ | 0.042 | 0.042 | 0.033 | 0.034 | 0.033 |
| | 0.001 * | 0.001 * | 0.002 * | 0.002 * | 0.003 * |
| Volatility persistence HF price $b_H$ | 0.393 | 0.503 | 0.487 | 0.354 | 0.377 |
| | 0.000 * | 0.000 * | 0.000 * | 0.000 * | 0.000 * |
| Spillover from HF price to stock returns $\delta_{H,S}$ | 1.394 | 1.409 | 1.093 | 1.167 | 1.153 |
| | 0.000 * | 0.000 * | 0.000 * | 0.000 * | 0.000 * |
| Asymmetric spillover effect from HF price to stock returns $\theta_{H,S}$ | 0.007 | −0.026 | −0.007 | 0.043 | 0.003 |
| | 0.781 | 0.595 | 0.730 | 0.424 | 0.817 |
| Correlation coefficient $\rho$ | −0.152 | −0.083 | −0.125 | −0.044 | −0.060 |
| **New Zealand** | **CFB-FE** | **CFB-T** | **CFB-HK** | **HFNV** | **HKP** |
| Volatility persistence stock returns $b_S$ | 0.778 | 0.777 | 0.778 | 0.778 | 0.778 |
| | 0.000 * | 0.000 * | 0.000 * | 0.000 * | 0.000 * |
| Spillover from stock returns to HF price $\delta_{S,H}$ | 0.102 | 0.093 | 0.099 | 0.111 | 0.093 |
| | 0.000 * | 0.000 * | 0.000 * | 0.000 * | 0.000 * |
| Asymmetric spillover effect from stock returns to HF price $\theta_{S,H}$ | 0.024 | 0.033 | 0.025 | 0.032 | 0.034 |
| | 0.027 ** | 0.006 * | 0.015 ** | 0.009 * | 0.001 * |
| Volatility persistence HF price $b_H$ | 0.381 | 0.415 | 0.475 | 0.432 | 0.408 |
| | 0.000 * | 0.000 * | 0.000 * | 0.000 * | 0.000 * |
| Spillover from HF price to stock returns $\delta_{H,S}$ | 1.026 | 1.232 | 1.334 | 1.259 | 1.173 |
| | 0.000 * | 0.000 * | 0.000 * | 0.000 * | 0.000 * |
| Asymmetric spillover effect from HF price to stock returns $\theta_{H,S}$ | 0.023 | 0.007 | −0.037 | −0.046 | 0.013 |
| | 0.503 | 0.709 | 0.327 | 0.307 | 0.655 * |
| Correlation coefficient $\rho$ | −0.161 | −0.110 | −0.161 | −0.007 | −0.085 |
| **New Zealand** | **IIF** | **IVI** | **IGF** | **JKAI** | **LIM** |
| Volatility persistence stock returns $b_S$ | 0.778 | 0.842 | 0.842 | 0.686 | 0.686 |
| | 0.000 * | 0.000 * | 0.000 * | 0.000 * | 0.000 * |
| Spillover from stock returns to HF price $\delta_{S,H}$ | 0.102 | 0.118 | 0.117 | 0.081 | 0.095 |
| | 0.000 * | 0.000 * | 0.000 * | 0.000 * | 0.000 * |
| Asymmetric spillover effect from stock returns to HF price $\theta_{S,H}$ | 0.032 | 0.024 | 0.026 | 0.038 | 0.022 |
| | 0.007 * | 0.045 ** | 0.032 ** | 0.000 * | 0.015 * |
| Volatility persistence HF price $b_H$ | 0.383 | 0.467 | 0.504 | 0.331 | 0.354 |
| | 0.000 * | 0.000 * | 0.000 * | 0.000 * | 0.000 * |
| Spillover from HF price to stock returns $\delta_{H,S}$ | 1.178 | 1.309 | 1.465 | 0.979 | 0.885 |
| | 0.000 * | 0.000 * | 0.000 * | 0.000 * | 0.000 * |
| Asymmetric spillover effect from HF price to stock returns $\theta_{H,S}$ | 0.026 | 0.022 | 0.022 | 0.006 | −0.008 |
| | 0.507 | 0.575 | 0.637 | 0.637 | 0.611 |
| Correlation coefficient $\rho$ | −0.067 | −0.064 | −0.124 | −0.037 | −0.134 |

**Table 3.** *Cont.*

| New Zealand | MLM | PPL | PIF | PCF | ISF |
|---|---|---|---|---|---|
| Volatility persistence stock returns $b_S$ | 0.687 | 0.687 | 0.687 | 0.687 | 0.686 |
| | 0.000 * | 0.000 * | 0.000 * | 0.000 * | 0.000 * |
| Spillover from stock returns to HF price $\delta_{S,H}$ | 0.087 | 0.075 | 0.082 | 0.081 | 0.079 |
| | 0.000 * | 0.000 * | 0.000 * | 0.000 * | 0.000 * |
| Asymmetric spillover effect from stock returns to HF price $\theta_{S,H}$ | 0.025 | 0.041 | 0.028 | 0.038 | 0.037 |
| | 0.003 * | 0.000 * | 0.001 * | 0.000 * | 0.000 * |
| Volatility persistence HF price $b_H$ | 0.278 | 0.305 | 0.300 | 0.375 | 0.340 |
| | 0.000 * | 0.000 * | 0.000 * | 0.000 * | 0.000 * |
| Spillover from HF price to stock returns $\delta_{H,S}$ | 0.907 | 1.025 | 1.066 | 1.213 | 0.958 |
| | 0.000 * | 0.000 * | 0.000 * | 0.000 * | 0.000 * |
| Asymmetric spillover effect from HF price to stock returns $\theta_{H,S}$ | −0.018 | −0.025 | 0.019 | 0.007 | −0.001 |
| | 0.508 | 0.454 | 0.521 | 0.631 | 0.684 |
| Correlation coefficient $\rho$ | −0.065 | −0.129 | −0.104 | −0.081 | −0.035 |
| **New Zealand** | **SJO** | **SRG** | **VPC-A** | **VPC-B** | **VEI** |
| Volatility persistence stock returns $b_S$ | 0.687 | 0.687 | 0.686 | 0.687 | 0.686 |
| | 0.000 * | 0.000 * | 0.000 * | 0.000 * | 0.000 * |
| Spillover from stock returns to HF price $\delta_{S,H}$ | 0.081 | 0.088 | 0.079 | 0.079 | 0.082 |
| | 0.000 * | 0.000 * | 0.000 * | 0.000 * | 0.000 * |
| Asymmetric spillover effect from stock returns to HF price $\theta_{S,H}$ | 0.029 | 0.028 | 0.032 | 0.032 | 0.027 |
| | 0.001 * | 0.002 * | 0.001 * | 0.000 * | 0.002 * |
| Volatility persistence HF price $b_H$ | 0.339 | 0.333 | 0.341 | 0.336 | 0.536 |
| | 0.000 * | 0.000 * | 0.000 * | 0.000 * | 0.000 * |
| Spillover from HF price to stock returns $\delta_{H,S}$ | 1.066 | 0.941 | 1.040 | 1.023 | 0.638 |
| | 0.000 * | 0.000 * | 0.000 * | 0.000 * | 0.000 * |
| Asymmetric spillover effect from HF price to stock returns $\theta_{H,S}$ | −0.044 | −0.015 | −0.028 | −0.028 | −0.007 |
| | 0.245 | 0.509 | 0.377 | 0.380 | 0.596 |
| Correlation coefficient $\rho$ | −0.045 | −0.051 | −0.105 | −0.106 | −0.002 |

* 1% significance level. ** 5% significance level.

Table 4 shows the results of volatility spillover between the Japanese stock market and the hedge funds market and vice versa. For volatility persistence, the coefficient (b) is significant for the Japan stock market. The volatility persistence for hedge funds is also significant. In addition, the values of ($b_S$) and ($b_H$) are less than one, a condition necessary to have stable volatility Wu (2005).

For volatility spillover, we find that the coefficient ($\delta$) is significant for the Japanese stock market impact on the hedge fund market but, for hedge fund market on the Japanese stock market, only few funds showed the spillover effect at the 5% significance level e.g., (BDP and VPC-B). In addition, HFNV, IIF, IVI, LIM, PPL, and PIF were significant at the 10% level. This implies the return spillover effect is not strong from hedge funds to the Japanese stock market. This knowledge of volatility spillover effects can be helpful in asset allocation and stock selection.

**Table 4.** The Volatility spillover between the Japanese Stock Market (JPY) and Hedge Fund Returns.

| Japan | AGF | AACF | AAGF | BDP | BAF |
|---|---|---|---|---|---|
| Volatility persistence stock returns $b_S$ | 0.023 | 0.023 | 0.032 | 0.013 | 0.034 |
| | 0.000 * | 0.000 * | 0.000 * | 0.000 * | 0.000 * |
| Spillover from stock returns to HF price $\delta_{S,H}$ | 0.010 | 0.010 | 0.011 | 0.004 | 0.010 |
| | 0.000 * | 0.000 * | 0.001 * | 0.000 * | 0.001 * |
| Asymmetric spillover effect from stock returns to HF price $\theta_{S,H}$ | 0.092 | 0.119 | 0.163 | 0.044 | 0.118 |
| | 0.000 * | 0.000 * | 0.000 * | 0.000 * | 0.000 * |
| Volatility persistence HF price $b_H$ | 0.328 | 0.332 | 0.366 | 0.144 | 0.360 |
| | 0.000 * | 0.000 * | 0.000 * | 0.000 * | 0.000 * |
| Spillover from HF price to stock returns $\delta_{H,S}$ | 0.002 | −0.006 | −0.002 | 0.005 | 0.001 |
| | 0.184 | 0.140 | 0.244 | 0.052 ** | 0.255 |
| Asymmetric spillover effect from HF price to stock returns $\theta_{H,S}$ | −0.036 | −0.019 | −0.042 | −0.005 | −0.019 |
| | 0.000 * | 0.000 * | 0.000 * | 0.000 * | 0.000 * |
| Correlation coefficient $\rho$ | −0.122 | −0.066 | −0.100 | −0.035 | −0.048 |
| **Japan** | **CFB-FE** | **CFB-T** | **CFB-HK** | **HFNV** | **HKP** |
| Volatility persistence stock returns $b_S$ | 0.031 | 0.029 | 0.031 | 0.034 | 0.029 |
| | 0.000 * | 0.000 * | 0.000 * | 0.000 * | 0.000 * |
| Spillover from stock returns to HF price $\delta_{S,H}$ | 0.007 | 0.010 | 0.008 | 0.010 | 0.011 |
| | 0.008 * | 0.002 * | 0.005 * | 0.003 * | 0.000 * |
| Asymmetric spillover effect from stock returns to HF price $\theta_{S,H}$ | 0.119 | 0.130 | 0.148 | 0.135 | 0.127 |
| | 0.000 * | 0.000 * | 0.000 * | 0.000 * | 0.000 * |
| Volatility persistence HF price $b_H$ | 0.320 | 0.384 | 0.416 | 0.393 | 0.366 |
| | 0.000 * | 0.000 * | 0.000 * | 0.000 * | 0.000 * |
| Spillover from HF price to stock returns $\delta_{H,S}$ | 0.007 | 0.002 | −0.011 | −0.014 | 0.004 |
| | 0.157 | 0.221 | 0.102 | 0.096 *** | 0.204 |
| Asymmetric spillover effect from HF price to stock returns $\theta_{H,S}$ | −0.050 | −0.034 | −0.050 | −0.002 | −0.027 |
| | 0.000 * | 0.000 * | 0.000 * | 0.000 * | 0.000 * |
| Correlation coefficient $\rho$ | −0.129 | −0.088 | −0.129 | −0.006 | −0.068 |
| **Japan** | **IIF** | **IVI** | **IGF** | **JKAI** | **LIM** |
| Volatility persistence stock returns $b_S$ | 0.016 | 0.019 | 0.019 | 0.013 | 0.015 |
| | 0.000 * | 0.000 * | 0.000 * | 0.000 * | 0.000 * |
| Spillover from stock returns to HF price $\delta_{S,H}$ | 0.005 | 0.004 | 0.004 | 0.006 | 0.004 |
| | 0.001 * | 0.007 * | 0.005 * | 0.000 * | 0.002 * |
| Asymmetric spillover effect from stock returns to HF price $\theta_{S,H}$ | 0.062 | 0.076 | 0.082 | 0.054 | 0.057 |
| | 0.000 * | 0.000 * | 0.000 * | 0.000 * | 0.000 * |
| Volatility persistence HF price $b_H$ | 0.191 | 0.212 | 0.237 | 0.159 | 0.143 |
| | 0.000 * | 0.000 * | 0.000 * | 0.000 * | 0.000 * |
| Spillover from HF price to stock returns $\delta_{H,S}$ | 0.004 | 0.004 | 0.004 | 0.001 | −0.001 |
| | 0.082 *** | 0.093 *** | 0.103 | 0.103 | 0.099 *** |
| Asymmetric spillover effect from HF price to stock returns $\theta_{H,S}$ | −0.011 | −0.010 | −0.020 | −0.006 | −0.022 |
| | 0.000 * | 0.000 * | 0.000 * | 0.000 * | 0.000 * |
| Correlation coefficient $\rho$ | −0.054 | −0.051 | −0.099 | −0.030 | −0.107 |

**Table 4.** *Cont.*

| Japan | MLM | PPL | PIF | PCF | ISF |
|---|---|---|---|---|---|
| Volatility persistence stock returns $b_S$ | 0.019 | 0.012 | 0.013 | 0.013 | 0.036 |
| | 0.000 * | 0.000 * | 0.000 * | 0.000 * | 0.000 * |
| Spillover from stock returns to HF price $\delta_{S,H}$ | 0.005 | 0.007 | 0.005 | 0.006 | 0.017 |
| | 0.001 * | 0.000 * | 0.000 * | 0.000 * | 0.000 * |
| Asymmetric spillover effect from stock returns to HF price $\theta_{S,H}$ | 0.060 | 0.049 | 0.049 | 0.061 | 0.154 |
| | 0.000 * | 0.000 * | 0.000 * | 0.000 * | 0.000 * |
| Volatility persistence HF price $b_H$ | 0.195 | 0.166 | 0.173 | 0.197 | 0.433 |
| | 0.000 * | 0.000 * | 0.000 * | 0.000 * | 0.000 * |
| Spillover from HF price to stock returns $\delta_{H,S}$ | −0.004 | −0.004 | 0.003 | 0.001 | 0.000 |
| | 0.109 | 0.074 *** | 0.084 *** | 0.102 | 0.309 |
| Asymmetric spillover effect from HF price to stock returns $\theta_{H,S}$ | −0.014 | −0.021 | −0.017 | −0.013 | −0.016 |
| | 0.000 * | 0.000 * | 0.000 * | 0.000 * | 0.000 * |
| Correlation coefficient $\rho$ | −0.052 | −0.103 | −0.083 | −0.065 | −0.028 |
| **Japan** | **SJO** | **SRG** | **VPC-A** | **VPC-B** | **VEI** |
| Volatility persistence stock returns $b_S$ | 0.037 | 0.040 | 0.036 | 0.009 | 0.036 |
| | 0.000 * | 0.000 * | 0.000 * | 0.000 * | 0.000 * |
| Spillover from stock returns to HF price $\delta_{S,H}$ | 0.013 | 0.013 | 0.014 | 0.004 | 0.012 |
| | 0.000 * | 0.001 * | 0.000 * | 0.000 * | 0.001 * |
| Asymmetric spillover effect from stock returns to HF price $\theta_{S,H}$ | 0.153 | 0.151 | 0.154 | 0.038 | 0.233 |
| | 0.000 * | 0.000 * | 0.000 * | 0.000 * | 0.000 * |
| Volatility persistence HF price $b_H$ | 0.482 | 0.425 | 0.470 | 0.115 | 0.277 |
| | 0.000 * | 0.000 * | 0.000 * | 0.000 * | 0.000 * |
| Spillover from HF price to stock returns $\delta_{H,S}$ | −0.020 | −0.007 | −0.012 | −0.003 | −0.003 |
| | 0.111 | 0.230 | 0.171 | 0.043 ** | 0.259 |
| Asymmetric spillover effect from HF price to stock returns $\theta_{H,S}$ | −0.021 | −0.023 | −0.048 | −0.012 | −0.001 |
| | 0.000 * | 0.000 * | 0.000 * | 0.000 * | 0.000 * |
| Correlation coefficient $\rho$ | −0.036 | −0.041 | −0.084 | −0.085 | −0.002 |

\* 1% significance level. ** 5% significance level. *** 10% significance level.

For the asymmetric spillover response, the coefficient ($\theta$) is significant for the Japanese stock market on the hedge funds market and the reciprocal is also significant. This implies that negative shocks from the Japanese stock market generate greater volatility in the hedge fund market than positive shocks of a similar magnitude. This also suggests that similar effects happen where negative shocks in the hedge funds market generate greater volatility in Japanese stock markets than positive shocks of a similar magnitude.

Table 5 shows the results of volatility analysis between the Singaporean stock market and the hedge funds market and vice versa. For volatility persistence, the coefficient (b) is significant for the Singaporean stock market. The volatility persistence for hedge funds is also significant. In addition, the values of ($b_S$) and ($b_H$) are less than one, a condition necessary to have stable volatility Wu (2005).

**Table 5.** The Volatility spillover between the Singapore Stock Market (STI) and Hedge Fund Returns.

| Singapore | AGF | AACF | AAGF | BDP | BAF |
|---|---|---|---|---|---|
| Volatility persistence stock returns $b_S$ | 0.997 | 0.998 | 0.997 | 0.998 | 0.995 |
|  | 0.000 * | 0.000 * | 0.000 * | 0.000 * | 0.000 * |
| Spillover from stock returns to HF price $\delta_{S,H}$ | 0.119 | 0.120 | 0.122 | 0.127 | 0.132 |
|  | 0.000 * | 0.000 * | 0.000 * | 0.000 * | 0.000 * |
| Asymmetric spillover effect from stock returns to HF price $\theta_{S,H}$ | 0.054 | 0.054 | 0.045 | 0.044 | 0.043 |
|  | 0.002 * | 0.001 * | 0.003 * | 0.003 * | 0.005 * |
| Volatility persistence HF price $b_H$ | 0.466 | 0.597 | 0.595 | 0.420 | 0.447 |
|  | 0.000 * | 0.000 * | 0.000 * | 0.000 * | 0.000 * |
| Spillover from HF price to stock returns $\delta_{H,S}$ | 1.650 | 1.668 | 1.334 | 1.383 | 1.366 |
|  | 0.000 * | 0.000 * | 0.000 * | 0.000 * | 0.000 * |
| Asymmetric spillover effect from HF price to stock returns $\theta_{H,S}$ | 0.008 | −0.031 | −0.008 | 0.052 | 0.004 |
|  | 0.936 | 0.716 | 0.902 | 0.513 | 0.979 |
| Correlation coefficient $\rho$ | −0.176 | −0.094 | −0.149 | −0.048 | −0.068 |
| **Singapore** | **CFB-FE** | **CFB-T** | **CFB-HK** | **HFNV** | **HKP** |
| Volatility persistence stock returns $b_S$ | 0.997 | 0.995 | 0.997 | 0.996 | 0.997 |
|  | 0.000 * | 0.000 * | 0.000 * | 0.000 * | 0.000 * |
| Spillover from stock returns to HF price $\delta_{S,H}$ | 0.132 | 0.122 | 0.129 | 0.144 | 0.122 |
|  | 0.000 * | 0.000 * | 0.000 * | 0.000 * | 0.000 * |
| Asymmetric spillover effect from stock returns to HF price $\theta_{S,H}$ | 0.034 | 0.046 | 0.037 | 0.044 | 0.048 |
|  | 0.035 ** | 0.009 * | 0.020 ** | 0.012 ** | 0.002 * |
| Volatility persistence HF price $b_H$ | 0.489 | 0.533 | 0.609 | 0.554 | 0.524 |
|  | 0.000 * | 0.000 * | 0.000 * | 0.000 * | 0.000 * |
| Spillover from HF price to stock returns $\delta_{H,S}$ | 1.315 | 1.578 | 1.710 | 1.613 | 1.503 |
|  | 0.000 * | 0.000 * | 0.000 * | 0.000 * | 0.000 * |
| Asymmetric spillover effect from HF price to stock returns $\theta_{H,S}$ | 0.030 | 0.009 | −0.046 | −0.058 | 0.018 |
|  | 0.656 | 0.920 | 0.430 | 0.405 | 0.851 |
| Correlation coefficient $\rho$ | −0.202 | −0.137 | −0.203 | −0.005 | −0.105 |
| **Singapore** | **IIF** | **IVI** | **IGF** | **JKAI** | **LIM** |
| Volatility persistence stock returns $b_S$ | 0.997 | 0.997 | 0.996 | 0.997 | 0.996 |
|  | 0.000 * | 0.000 * | 0.000 * | 0.000 * | 0.000 * |
| Spillover from stock returns to HF price $\delta_{S,H}$ | 0.133 | 0.141 | 0.140 | 0.120 | 0.140 |
|  | 0.000 * | 0.000 * | 0.000 * | 0.000 * | 0.000 * |
| Asymmetric spillover effect from stock returns to HF price $\theta_{S,H}$ | 0.045 | 0.032 | 0.035 | 0.059 | 0.036 |
|  | 0.009 * | 0.053 ** | 0.039 * | 0.001 * | 0.022 ** |
| Volatility persistence HF price $b_H$ | 0.491 | 0.554 | 0.598 | 0.481 | 0.515 |
|  | 0.000 * | 0.000 * | 0.000 * | 0.000 * | 0.000 * |
| Spillover from HF price to stock returns $\delta_{H,S}$ | 1.509 | 1.550 | 1.734 | 1.422 | 1.285 |
|  | 0.000 * | 0.000 * | 0.000 * | 0.000 * | 0.000 * |
| Asymmetric spillover effect from HF price to stock returns $\theta_{H,S}$ | 0.034 | 0.027 | 0.027 | 0.009 | −0.011 |
|  | 0.662 | 0.693 | 0.766 | 0.937 | 0.899 |
| Correlation coefficient $\rho$ | −0.082 | −0.071 | −0.143 | −0.050 | −0.191 |

**Table 5.** *Cont.*

| Singapore | MLM | PPL | PIF | PCF | ISF |
|---|---|---|---|---|---|
| Volatility persistence stock returns $b_S$ | 0.997 | 0.997 | 0.998 | 0.997 | 0.997 |
| | 0.000 * | 0.000 * | 0.000 * | 0.000 * | 0.000 * |
| Spillover from stock returns to HF price $\delta_{S,H}$ | 0.128 | 0.111 | 0.121 | 0.120 | 0.117 |
| | 0.000 * | 0.000 * | 0.000 * | 0.000 * | 0.000 * |
| Asymmetric spillover effect from stock returns to HF price $\theta_{S,H}$ | 0.040 | 0.063 | 0.045 | 0.058 | 0.058 |
| | 0.005 * | 0.001 * | 0.002 * | 0.001 * | 0.001 * |
| Volatility persistence HF price $b_H$ | 0.404 | 0.444 | 0.437 | 0.545 | 0.495 |
| | 0.000 * | 0.000 * | 0.000 * | 0.000 * | 0.000 * |
| Spillover from HF price to stock returns $\delta_{H,S}$ | 1.318 | 1.489 | 1.548 | 1.762 | 1.391 |
| | 0.000 * | 0.000 * | 0.000 * | 0.000 * | 0.000 * |
| Asymmetric spillover effect from HF price to stock returns $\theta_{H,S}$ | −0.026 | −0.036 | 0.029 | 0.010 | 0.000 |
| | 0.749 | 0.671 | 0.769 | 0.927 | 1.005 |
| Correlation coefficient $\rho$ | −0.090 | −0.184 | −0.146 | −0.113 | −0.047 |
| **Singapore** | **SJO** | **SRG** | **VPC-A** | **VPC-B** | **VEI** |
| Volatility persistence stock returns $b_S$ | 0.997 | 0.997 | 0.996 | 0.997 | 0.996 |
| | 0.000 * | 0.000 * | 0.000 * | 0.000 * | 0.000 * |
| Spillover from stock returns to HF price $\delta_{S,H}$ | 0.119 | 0.130 | 0.117 | 0.117 | 0.121 |
| | 0.000 * | 0.000 * | 0.000 * | 0.000 * | 0.000 * |
| Asymmetric spillover effect from stock returns to HF price $\theta_{S,H}$ | 0.046 | 0.045 | 0.050 | 0.051 | 0.043 |
| | 0.002 * | 0.003 * | 0.001 * | 0.001 * | 0.004 * |
| Volatility persistence HF price $b_H$ | 0.493 | 0.484 | 0.496 | 0.489 | 0.779 |
| | 0.000 * | 0.000 * | 0.000 * | 0.000 * | 0.000 * |
| Spillover from HF price to stock returns $\delta_{H,S}$ | 1.548 | 1.367 | 1.511 | 1.486 | 0.927 |
| | 0.000 * | 0.000 * | 0.000 * | 0.000 * | 0.000 * |
| Asymmetric spillover effect from HF price to stock returns $\theta_{H,S}$ | −0.063 | −0.021 | −0.040 | −0.040 | −0.009 |
| | 0.367 | 0.751 | 0.560 | 0.563 | 0.877 |
| Correlation coefficient $\rho$ | −0.062 | −0.070 | −0.149 | −0.150 | 0.001 |

\* 1% significance level. \*\* 5% significance level.

For volatility spillover, we find that the coefficient ($\delta$) is again significant, which suggests bidirectional volatility spillover exists between the Singaporean stock market and the hedge funds market. For the asymmetric spillover response, the coefficient ($\theta$) is also significant for the Singaporean stock market on the hedge funds market and the reciprocal is also significant. This implies that negative shocks from the Singaporean stock market generate greater volatility in the hedge fund market than positive shocks of a similar magnitude.

Table 6 shows the results of the volatility analysis of the Indian stock market and the hedge fund market and vice versa. For volatility persistence, the coefficient (b) is significant for the India stock market. The volatility persistence for hedge funds is also significant. In addition, the values of ($b_S$) and ($b_H$) are less than one, a condition necessary to have stable volatility Wu (2005). For volatility spillover, we find that the coefficient ($\delta$) is again significant, which suggests the bidirectional volatility spillover exists between the Indian stock market and the hedge fund market. For the asymmetric spillover response, the coefficient ($\theta$) is also significant for the Indian stock market on the hedge funds market and the reciprocal is also significant. This implies that negative shocks from the Indian stock market generate greater volatility in the hedge fund market than positive shocks of a similar magnitude.

**Table 6.** Volatility Spillover Between the Indian Stock Market (NSE500) and Hedge Fund Returns.

| India | AGF | AACF | AAGF | BDP | BAF |
|---|---|---|---|---|---|
| Volatility persistence stock returns $b_S$ | 0.998 | 0.998 | 0.998 | 0.998 | 0.996 |
| | 0.000 * | 0.000 * | 0.000 * | 0.000 * | 0.000 * |
| Spillover from stock returns to HF price $\delta_{S,H}$ | 0.119 | 0.120 | 0.122 | 0.127 | 0.132 |
| | 0.000 * | 0.000 * | 0.000 * | 0.000 * | 0.000 * |
| Asymmetric spillover effect from stock returns to HF price $\theta_{S,H}$ | 0.053 | 0.053 | 0.044 | 0.043 | 0.042 |
| | 0.002 * | 0.002 * | 0.003 * | 0.003 * | 0.005 * |
| Volatility persistence HF price $b_H$ | 0.546 | 0.677 | 0.675 | 0.500 | 0.527 |
| | 0.000 * | 0.000 * | 0.000 * | 0.000 * | 0.000 * |
| Spillover from HF price to stock returns $\delta_{H,S}$ | 1.650 | 1.668 | 1.334 | 1.383 | 1.366 |
| | 0.000 * | 0.000 * | 0.000 * | 0.000 * | 0.000 * |
| Asymmetric spillover effect from HF price to stock returns $\theta_{H,S}$ | 0.011 | −0.028 | −0.005 | 0.054 | 0.006 |
| | 0.928 | 0.708 | 0.894 | 0.505 | 0.971 |
| Correlation coefficient $\rho$ | −0.174 | −0.092 | −0.147 | −0.046 | −0.066 |
| **India** | **CFB-FE** | **CFB-T** | **CFB-HK** | **HFNV** | **HKP** |
| Volatility persistence stock returns $b_S$ | 0.997 | 0.996 | 0.997 | 0.997 | 0.997 |
| | 0.000 * | 0.000 * | 0.000 * | 0.000 * | 0.000 * |
| Spillover from stock returns to HF price $\delta_{S,H}$ | 0.132 | 0.122 | 0.129 | 0.144 | 0.122 |
| | 0.000 * | 0.000 * | 0.000 * | 0.000 * | 0.000 * |
| Asymmetric spillover effect from stock returns to HF price $\theta_{S,H}$ | 0.033 | 0.045 | 0.036 | 0.043 | 0.047 |
| | 0.036 ** | 0.009 * | 0.021 ** | 0.012 ** | 0.002 * |
| Volatility persistence HF price $b_H$ | 0.569 | 0.613 | 0.689 | 0.634 | 0.604 |
| | 0.000 * | 0.000 * | 0.000 * | 0.000 * | 0.000 * |
| Spillover from HF price to stock returns $\delta_{H,S}$ | 1.315 | 1.578 | 1.710 | 1.613 | 1.503 |
| | 0.000 * | 0.000 * | 0.000 * | 0.000 * | 0.000 * |
| Asymmetric spillover effect from HF price to stock returns $\theta_{H,S}$ | 0.033 | 0.011 | −0.044 | −0.056 | 0.020 |
| | 0.648 | 0.912 | 0.422 | 0.397 | 0.843 |
| Correlation coefficient $\rho$ | −0.200 | −0.135 | −0.201 | −0.003 | −0.103 |
| **India** | **IIF** | **IVI** | **IGF** | **JKAI** | **LIM** |
| Volatility persistence stock returns $b_S$ | 0.998 | 0.997 | 0.997 | 0.997 | 0.997 |
| | 0.000 * | 0.000 * | 0.000 * | 0.000 * | 0.000 * |
| Spillover from stock returns to HF price $\delta_{S,H}$ | 0.133 | 0.141 | 0.140 | 0.120 | 0.140 |
| | 0.000 * | 0.000 * | 0.000 * | 0.000 * | 0.000 * |
| Asymmetric spillover effect from stock returns to HF price $\theta_{S,H}$ | 0.044 | 0.031 | 0.034 | 0.058 | 0.035 |
| | 0.010 ** | 0.054 *** | 0.039 ** | 0.001 * | 0.022 ** |
| Volatility persistence HF price $b_H$ | 0.571 | 0.634 | 0.678 | 0.561 | 0.595 |
| | 0.000 * | 0.000 * | 0.000 * | 0.000 * | 0.000 * |
| Spillover from HF price to stock returns $\delta_{H,S}$ | 1.509 | 1.550 | 1.734 | 1.422 | 1.285 |
| | 0.000 * | 0.000 * | 0.000 * | 0.000 * | 0.000 * |
| Asymmetric spillover effect from HF price to stock returns $\theta_{H,S}$ | 0.036 | 0.029 | 0.029 | 0.011 | −0.009 |
| | 0.654 | 0.685 | 0.758 | 0.929 | 0.891 |
| Correlation coefficient $\rho$ | −0.080 | −0.069 | −0.141 | −0.048 | −0.189 |

**Table 6.** *Cont.*

| India | MLM | PPL | PIF | PCF | ISF |
|---|---|---|---|---|---|
| Volatility persistence stock returns $b_S$ | 0.997 | 0.998 | 0.998 | 0.998 | 0.997 |
| | 0.000 * | 0.000 * | 0.000 * | 0.000 * | 0.000 * |
| Spillover from stock returns to HF price $\delta_{S,H}$ | 0.128 | 0.111 | 0.121 | 0.120 | 0.117 |
| | 0.000 * | 0.000 * | 0.000 * | 0.000 * | 0.000 * |
| Asymmetric spillover effect from stock returns to HF price $\theta_{S,H}$ | 0.039 | 0.062 | 0.044 | 0.057 | 0.057 |
| | 0.005 * | 0.001 * | 0.002 * | 0.001 * | 0.001 * |
| Volatility persistence HF price $b_H$ | 0.484 | 0.524 | 0.517 | 0.625 | 0.575 |
| | 0.000 * | 0.000 * | 0.000 * | 0.000 * | 0.000 * |
| Spillover from HF price to stock returns $\delta_{H,S}$ | 1.318 | 1.489 | 1.548 | 1.762 | 1.391 |
| | 0.000 * | 0.000 * | 0.000 * | 0.000 * | 0.000 * |
| Asymmetric spillover effect from HF price to stock returns $\theta_{H,S}$ | −0.024 | −0.034 | 0.031 | 0.013 | 0.002 |
| | 0.741 | 0.663 | 0.761 | 0.919 | 0.997 |
| Correlation coefficient $\rho$ | −0.088 | −0.182 | −0.144 | −0.111 | −0.045 |
| **India** | **SJO** | **SRG** | **VPC-A** | **VPC-B** | **VEI** |
| Volatility persistence stock returns $b_S$ | 0.998 | 0.997 | 0.997 | 0.997 | 0.997 |
| | 0.000 * | 0.000 * | 0.000 * | 0.000 * | 0.000 * |
| Spillover from stock returns to HF price $\delta_{S,H}$ | 0.119 | 0.130 | 0.117 | 0.117 | 0.121 |
| | 0.000 * | 0.000 * | 0.000 * | 0.000 * | 0.000 * |
| Asymmetric spillover effect from stock returns to HF price $\theta_{S,H}$ | 0.045 | 0.044 | 0.049 | 0.050 | 0.042 |
| | 0.002 * | 0.003 * | 0.002 * | 0.001 * | 0.004 * |
| Volatility persistence HF price $b_H$ | 0.573 | 0.564 | 0.576 | 0.569 | 0.859 |
| | 0.000 * | 0.000 * | 0.000 * | 0.000 * | 0.000 * |
| Spillover from HF price to stock returns $\delta_{H,S}$ | 1.548 | 1.367 | 1.511 | 1.486 | 0.927 |
| | 0.000 * | 0.000 * | 0.000 * | 0.000 * | 0.000 * |
| Asymmetric spillover effect from HF price to stock returns $\theta_{H,S}$ | −0.061 | −0.018 | −0.037 | −0.038 | −0.007 |
| | 0.359 | 0.743 | 0.552 | 0.555 | 0.869 |
| Correlation coefficient $\rho$ | −0.060 | −0.068 | −0.147 | −0.148 | 0.003 |

* 1% significance level. ** 5% significance level. *** 10% significance level.

Table 7 shows the results of the impact of the South Korean stock market on the hedge fund market and vice versa. For volatility persistence, the coefficient (b) is significant for the South Korean stock market. The volatility persistence for hedge funds on South Korean stock market is also significant. In addition, the values of $b_S$ are greater than one. This suggests an unstable volatility spillover from the South Korean stock market returns on hedge fund returns. For hedge fund, ($b_H$) is less than one, like the Indian stock market, a condition necessary to have stable volatility Wu (2005).

For volatility spillover, we find that coefficient (δ) is again significant, which suggests bidirectional volatility spillover exists between the South Korean stock market and the hedge fund market. For the asymmetric spillover response, the coefficient (θ) is also significant for the South Korean stock market on the hedge fund market, but the reciprocal is insignificant. This implies that negative shocks from the South Korean stock market generate greater volatility in hedge fund market than positive shocks of a similar magnitude.

**Table 7.** Volatility spillover between the South Korean Stock Market (KOSPI) and Hedge Fund Returns.

| South Korea | AGF | AACF | AAGF | BDP | BAF |
|---|---|---|---|---|---|
| Volatility persistence stock returns $b_S$ | 1.048 | 1.048 | 1.048 | 1.048 | 1.046 |
| | 0.000 * | 0.000 * | 0.000 * | 0.000 * | 0.000 * |
| Spillover from stock returns to HF price $\delta_{S,H}$ | 0.155 | 0.156 | 0.158 | 0.163 | 0.168 |
| | 0.000 * | 0.000 * | 0.000 * | 0.000 * | 0.000 * |
| Asymmetric spillover effect from stock returns to HF price $\theta_{S,H}$ | 0.098 | 0.098 | 0.089 | 0.088 | 0.087 |
| | 0.009 * | 0.009 * | 0.010 * | 0.010 ** | 0.012 ** |
| Volatility persistence HF price $b_H$ | 0.550 | 0.681 | 0.679 | 0.504 | 0.531 |
| | 0.000 * | 0.000 * | 0.000 * | 0.000 * | 0.000 * |
| Spillover from HF price to stock returns $\delta_{H,S}$ | 1.652 | 1.670 | 1.336 | 1.385 | 1.368 |
| | 0.000 * | 0.000 * | 0.000 * | 0.000 * | 0.000 * |
| Asymmetric spillover effect from HF price to stock returns $\theta_{H,S}$ | 0.018 | −0.021 | 0.002 | 0.061 | 0.013 |
| | 0.936 | 0.716 | 0.902 | 0.513 | 0.979 |
| Correlation coefficient $\rho$ | −0.168 | −0.086 | −0.141 | −0.040 | −0.060 |
| **South Korea** | **CFB-FE** | **CFB-T** | **CFB-HK** | **HFNV** | **HKP** |
| Volatility persistence stock returns $b_S$ | 1.047 | 1.046 | 1.047 | 1.047 | 1.047 |
| | 0.000 * | 0.000 * | 0.000 * | 0.000 * | 0.000 * |
| Spillover from stock returns to HF price $\delta_{S,H}$ | 0.168 | 0.158 | 0.165 | 0.180 | 0.158 |
| | 0.000 * | 0.000 * | 0.000 * | 0.000 * | 0.000 * |
| Asymmetric spillover effect from stock returns to HF price $\theta_{S,H}$ | 0.078 | 0.090 | 0.081 | 0.088 | 0.092 |
| | 0.043 ** | 0.016 ** | 0.028 ** | 0.019 ** | 0.009 * |
| Volatility persistence HF price $b_H$ | 0.573 | 0.617 | 0.693 | 0.638 | 0.608 |
| | 0.000 * | 0.000 * | 0.000 * | 0.000 * | 0.000 * |
| Spillover from HF price to stock returns $\delta_{H,S}$ | 1.317 | 1.580 | 1.712 | 1.615 | 1.505 |
| | 0.000 * | 0.000 * | 0.000 * | 0.000 * | 0.000 * |
| Asymmetric spillover effect from HF price to stock returns $\theta_{H,S}$ | 0.040 | 0.018 | −0.037 | −0.049 | 0.027 |
| | 0.656 | 0.920 | 0.430 | 0.405 | 0.851 |
| Correlation coefficient $\rho$ | −0.194 | −0.129 | −0.195 | 0.003 | −0.097 |
| **South Korea** | **IIF** | **IVI** | **IGF** | **JKAI** | **LIM** |
| Volatility persistence stock returns $b_S$ | 1.048 | 1.047 | 1.047 | 1.047 | 1.047 |
| | 0.000 * | 0.000 * | 0.000 * | 0.000 * | 0.000 * |
| Spillover from stock returns to HF price $\delta_{S,H}$ | 0.169 | 0.177 | 0.176 | 0.156 | 0.176 |
| | 0.000 * | 0.000 * | 0.000 * | 0.000 * | 0.000 * |
| Asymmetric spillover effect from stock returns to HF price $\theta_{S,H}$ | 0.089 | 0.076 | 0.079 | 0.103 | 0.080 |
| | 0.017 ** | 0.061 *** | 0.046 ** | 0.008 * | 0.029 ** |
| Volatility persistence HF price $b_H$ | 0.575 | 0.638 | 0.682 | 0.565 | 0.599 |
| | 0.000 * | 0.000 * | 0.000 * | 0.000 * | 0.000 * |
| Spillover from HF price to stock returns $\delta_{H,S}$ | 1.511 | 1.552 | 1.736 | 1.424 | 1.287 |
| | 0.000 * | 0.000 * | 0.000 * | 0.000 * | 0.000 * |
| Asymmetric spillover effect from HF price to stock returns $\theta_{H,S}$ | 0.043 | 0.036 | 0.036 | 0.018 | −0.002 |
| | 0.662 | 0.693 | 0.766 | 0.937 | 0.899 |
| Correlation coefficient $\rho$ | −0.074 | −0.063 | −0.135 | −0.042 | −0.183 |

**Table 7.** *Cont.*

| South Korea | MLM | PPL | PIF | PCF | ISF |
|---|---|---|---|---|---|
| Volatility persistence stock returns $b_S$ | 1.047 | 1.048 | 1.048 | 1.048 | 1.047 |
| | 0.000 * | 0.000 * | 0.000 * | 0.000 * | 0.000 * |
| Spillover from stock returns to HF price $\delta_{S,H}$ | 0.164 | 0.147 | 0.157 | 0.156 | 0.153 |
| | 0.000 * | 0.000 * | 0.000 * | 0.000 * | 0.000 * |
| Asymmetric spillover effect from stock returns to HF price $\theta_{S,H}$ | 0.084 | 0.107 | 0.089 | 0.102 | 0.102 |
| | 0.012 ** | 0.008 * | 0.009 * | 0.008 * | 0.008 * |
| Volatility persistence HF price $b_H$ | 0.488 | 0.528 | 0.521 | 0.629 | 0.579 |
| | 0.000 * | 0.000 * | 0.000 * | 0.000 * | 0.000 * |
| Spillover from HF price to stock returns $\delta_{H,S}$ | 1.320 | 1.491 | 1.550 | 1.764 | 1.393 |
| | 0.000 * | 0.000 * | 0.000 * | 0.000 * | 0.000 * |
| Asymmetric spillover effect from HF price to stock returns $\theta_{H,S}$ | −0.017 | −0.027 | 0.038 | 0.020 | 0.009 |
| | 0.749 | 0.671 | 0.769 | 0.927 | 1.005 |
| Correlation coefficient $\rho$ | −0.082 | −0.176 | −0.138 | −0.105 | −0.039 |
| **South Korea** | **SJO** | **SRG** | **VPC-A** | **VPC-B** | **VEI** |
| Volatility persistence stock returns $b_S$ | 1.048 | 1.047 | 1.047 | 1.047 | 1.047 |
| | 0.000 * | 0.000 * | 0.000 * | 0.000 * | 0.000 * |
| Spillover from stock returns to HF price $\delta_{S,H}$ | 0.155 | 0.166 | 0.153 | 0.153 | 0.157 |
| | 0.000 * | 0.000 * | 0.000 * | 0.000 * | 0.000 * |
| Asymmetric spillover effect from stock returns to HF price $\theta_{S,H}$ | 0.090 | 0.089 | 0.094 | 0.095 | 0.087 |
| | 0.009 * | 0.010 ** | 0.009 * | 0.008 * | 0.011 ** |
| Volatility persistence HF price $b_H$ | 0.577 | 0.568 | 0.580 | 0.573 | 0.863 |
| | 0.000 * | 0.000 * | 0.000 * | 0.000 * | 0.000 * |
| Spillover from HF price to stock returns $\delta_{H,S}$ | 1.550 | 1.369 | 1.513 | 1.488 | 0.929 |
| | 0.000 * | 0.000 * | 0.000 * | 0.000 * | 0.000 * |
| Asymmetric spillover effect from HF price to stock returns $\theta_{H,S}$ | −0.054 | −0.011 | −0.030 | −0.031 | 0.000 |
| | 0.367 | 0.751 | 0.560 | 0.563 | 0.877 |
| Correlation coefficient $\rho$ | −0.054 | −0.062 | −0.141 | −0.142 | 0.009 |

\* 1% significance level, \*\* 5% significance level, \*\*\* 10% significance level.

Table 8 shows the results of volatility analysis between the Chinese stock market and the hedge funds market and vice versa. For volatility persistence, the coefficient (b) is significant for the Chinese stock market. The volatility persistence for hedge funds on China is also significant. In addition, the values of ($b_S$) and ($b_H$) are less than one, like the Indian and South Korean stock markets, a condition necessary to have stable volatility Wu (2005). In other words, the volatility spillover will have long term impact.

For volatility spillover, we find that coefficient (δ) is again significant, which suggests bidirectional volatility spillover exists between the Chinese stock market and the hedge fund market. For the asymmetric spillover response, the coefficient (θ) is also significant for the Chinese stock market on the hedge fund market, but the reciprocal is insignificant. The asymmetric spillover response coefficient (θ) has negative *p*-value and is significant at the 10% level. This suggests that the Chinese stock market has experienced a negative shock or received bad news that could cause the conditional variance of the hedge fund market returns to become more volatile and riskier. This implies that negative shocks from the Chinese stock market generate greater volatility in the hedge fund markets than positive shocks of a similar magnitude.

**Table 8.** Volatility spillover between the Chinese Stock Market (SHCOMP) and Hedge Fund Returns.

| China | AGF | AACF | AAGF | BDP | BAF |
|---|---|---|---|---|---|
| Volatility persistence stock returns $b_S$ | 0.948 | 0.948 | 0.948 | 0.948 | 0.946 |
| | 0.000 * | 0.000 * | 0.000 * | 0.000 * | 0.000 * |
| Spillover from stock returns to HF price $\delta_{S,H}$ | 0.055 | 0.056 | 0.058 | 0.063 | 0.068 |
| | 0.100 * | 0.100 * | 0.100 * | 0.100 * | 0.100 * |
| Asymmetric spillover effect from stock returns to HF price $\theta_{S,H}$ | −0.002 | −0.002 | −0.011 | −0.012 | −0.013 |
| | 0.091 *** | 0.092 *** | 0.090 *** | 0.090 *** | 0.088 *** |
| Volatility persistence HF price $b_H$ | 0.450 | 0.581 | 0.579 | 0.404 | 0.431 |
| | 0.000 * | 0.000 * | 0.000 * | 0.000 * | 0.000 * |
| Spillover from HF price to stock returns $\delta_{H,S}$ | 1.552 | 1.570 | 1.236 | 1.285 | 1.268 |
| | 0.000 * | 0.000 * | 0.000 * | 0.000 * | 0.000 * |
| Asymmetric spillover effect from HF price to stock returns $\theta_{H,S}$ | −0.082 | −0.121 | −0.098 | −0.039 | −0.087 |
| | 0.836 | 0.616 | 0.802 | 0.413 | 0.879 |
| Correlation coefficient $\rho$ | −0.268 | −0.186 | −0.241 | −0.140 | −0.160 |
| **China** | **CFB-FE** | **CFB-T** | **CFB-HK** | **HFNV** | **HKP** |
| Volatility persistence stock returns $b_S$ | 0.947 | 0.946 | 0.947 | 0.947 | 0.947 |
| | 0.000 * | 0.000 * | 0.000 * | 0.000 * | 0.000 * |
| Spillover from stock returns to HF price $\delta_{S,H}$ | 0.068 | 0.058 | 0.065 | 0.080 | 0.058 |
| | 0.100 * | 0.100 * | 0.100 * | 0.100 * | 0.100 * |
| Asymmetric spillover effect from stock returns to HF price $\theta_{S,H}$ | −0.022 | −0.010 | −0.019 | −0.012 | −0.008 |
| | 0.058 ** | 0.084 *** | 0.073 *** | 0.081 *** | 0.091 *** |
| Volatility persistence HF price $b_H$ | 0.473 | 0.517 | 0.593 | 0.538 | 0.508 |
| | 0.000 * | 0.000 * | 0.000 * | 0.000 * | 0.000 * |
| Spillover from HF price to stock returns $\delta_{H,S}$ | 1.217 | 1.480 | 1.612 | 1.515 | 1.405 |
| | 0.000 * | 0.000 * | 0.000 * | 0.000 * | 0.000 * |
| Asymmetric spillover effect from HF price to stock returns $\theta_{H,S}$ | −0.060 | −0.082 | −0.137 | −0.149 | −0.073 |
| | 0.556 | 0.820 | 0.330 | 0.305 | 0.751 |
| Correlation coefficient $\rho$ | −0.294 | −0.229 | −0.295 | −0.097 | −0.197 |
| **China** | **IIF** | **IVI** | **IGF** | **JKAI** | **LIM** |
| Volatility persistence stock returns $b_S$ | 0.948 | 0.947 | 0.947 | 0.947 | 0.947 |
| | 0.000 * | 0.000 * | 0.000 * | 0.000 * | 0.000 * |
| Spillover from stock returns to HF price $\delta_{S,H}$ | 0.069 | 0.077 | 0.076 | 0.056 | 0.076 |
| | 0.100 * | 0.100 * | 0.100 * | 0.100 * | 0.100 * |
| Asymmetric spillover effect from stock returns to HF price $\theta_{S,H}$ | −0.011 | −0.024 | −0.021 | 0.003 | −0.020 |
| | 0.083 *** | 0.040 *** | 0.054 *** | 0.092 *** | 0.071 *** |
| Volatility persistence HF price $b_H$ | 0.475 | 0.538 | 0.582 | 0.465 | 0.499 |
| | 0.000 * | 0.000 * | 0.000 * | 0.000 * | 0.000 * |
| Spillover from HF price to stock returns $\delta_{H,S}$ | 1.411 | 1.452 | 1.636 | 1.324 | 1.187 |
| | 0.000 * | 0.000 * | 0.000 * | 0.000 * | 0.000 * |
| Asymmetric spillover effect from HF price to stock returns $\theta_{H,S}$ | −0.057 | −0.064 | −0.064 | −0.082 | −0.102 |
| | 0.562 | 0.593 | 0.666 | 0.837 | 0.799 |
| Correlation coefficient $\rho$ | −0.174 | −0.163 | −0.235 | −0.142 | −0.283 |

**Table 8.** *Cont.*

| China | MLM | PPL | PIF | PCF | ISF |
|---|---|---|---|---|---|
| Volatility persistence stock returns $b_S$ | 0.947 | 0.948 | 0.948 | 0.948 | 0.947 |
| | 0.000 * | 0.000 * | 0.000 * | 0.000 * | 0.000 * |
| Spillover from stock returns to HF price $\delta_{S,H}$ | 0.064 | 0.047 | 0.057 | 0.056 | 0.053 |
| | 0.100 * | 0.100 * | 0.100 * | 0.100 * | 0.100 * |
| Asymmetric spillover effect from stock returns to HF price $\theta_{S,H}$ | −0.016 | 0.007 | −0.011 | 0.002 | 0.002 |
| | 0.088 *** | 0.092 *** | 0.091 *** | 0.092 *** | 0.092 *** |
| Volatility persistence HF price $b_H$ | 0.388 | 0.428 | 0.421 | 0.529 | 0.479 |
| | 0.000 * | 0.000 * | 0.000 * | 0.000 * | 0.000 * |
| Spillover from HF price to stock returns $\delta_{H,S}$ | 1.220 | 1.391 | 1.450 | 1.664 | 1.293 |
| | 0.000 * | 0.000 * | 0.000 * | 0.000 * | 0.000 * |
| Asymmetric spillover effect from HF price to stock returns $\theta_{H,S}$ | −0.117 | −0.127 | −0.062 | −0.080 | −0.091 |
| | 0.649 | 0.571 | 0.669 | 0.827 | 0.905 |
| Correlation coefficient $\rho$ | −0.182 | −0.276 | −0.238 | −0.205 | −0.139 |
| **China** | **SJO** | **SRG** | **VPC-A** | **VPC-B** | **VEI** |
| Volatility persistence stock returns $b_S$ | 0.948 | 0.947 | 0.947 | 0.947 | 0.947 |
| | 0.000 * | 0.000 * | 0.000 * | 0.000 * | 0.000 * |
| Spillover from stock returns to HF price $\delta_{S,H}$ | 0.055 | 0.066 | 0.053 | 0.053 | 0.057 |
| | 0.100 * | 0.100 * | 0.100 * | 0.100 * | 0.100 * |
| Asymmetric spillover effect from stock returns to HF price $\theta_{S,H}$ | −0.010 | −0.011 | −0.006 | −0.005 | −0.013 |
| | 0.091 *** | 0.090 *** | 0.091 *** | 0.092 *** | 0.089 *** |
| Volatility persistence HF price $b_H$ | 0.477 | 0.468 | 0.480 | 0.473 | 0.763 |
| | 0.000 * | 0.000 * | 0.000 * | 0.000 * | 0.000 * |
| Spillover from HF price to stock returns $\delta_{H,S}$ | 1.450 | 1.269 | 1.413 | 1.388 | 0.829 |
| | 0.000 * | 0.000 * | 0.000 * | 0.000 * | 0.000 * |
| Asymmetric spillover effect from HF price to stock returns $\theta_{H,S}$ | −0.154 | −0.111 | −0.130 | −0.131 | −0.100 |
| | 0.267 | 0.651 | 0.460 | 0.463 | 0.777 |
| Correlation coefficient $\rho$ | −0.154 | −0.162 | −0.241 | −0.242 | −0.091 |

* 1% significance level, *** 10% significance level.

Table 9 shows the results of volatility analysis between the Hong Kong stock market and the hedge fund market and vice versa. For volatility persistence, the coefficient (b) is significant for the Hong Kong stock market. Volatility persistence for the hedge fund market on the Hong Kong stock market is significant. In addition, the values of ($b_S$) and ($b_H$) are less than one like most Asia-Pacific stock markets under study, a condition necessary to have stable volatility Wu (2005). In other words, volatility spillover will have long term impact.

For volatility spillover, the coefficient (δ) is significant, which suggests bidirectional volatility spillover exists between the Hong Kong stock market and the hedge fund market. For the asymmetric spillover response, the coefficient (θ) is also significant for the Hong Kong stock market on the hedge fund market at the 10% significance level, but the reciprocal is insignificant. For the asymmetric spillover response, the coefficient (θ) has a negative *p*-value and is significant at the 10% level. This suggests that if the Hong Kong stock market experienced a negative shock or received bad news, it could cause the conditional variance of hedge fund market returns to become more volatile and riskier. This implies that negative shocks from the Hong Kong stock market generate greater volatility in the hedge fund market than positive shocks of a similar magnitude.

**Table 9.** Volatility spillover between the Hong Kong Stock Market (HSI) and Hedge Fund Returns.

| Hong Kong | AGF | AACF | AAGF | BDP | BAF |
|---|---|---|---|---|---|
| Volatility persistence stock returns $b_S$ | 0.948 | 0.948 | 0.948 | 0.947 | 0.946 |
| | 0.000 * | 0.000 * | 0.000 * | 0.000 * | 0.000 * |
| Spillover from stock returns to HF price $\delta_{S,H}$ | 0.055 | 0.055 | 0.058 | 0.062 | 0.068 |
| | 0.100 *** | 0.100 *** | 0.100 *** | 0.100 *** | 0.100 *** |
| Asymmetric spillover effect from stock returns to HF price $\theta_{S,H}$ | −0.002 | −0.002 | −0.012 | −0.012 | −0.013 |
| | 0.091 *** | 0.092 *** | 0.090 *** | 0.090 *** | 0.089 *** |
| Volatility persistence HF price $b_H$ | 0.450 | 0.580 | 0.579 | 0.404 | 0.431 |
| | 0.000 * | 0.000 * | 0.000 * | 0.000 * | 0.000 * |
| Spillover from HF price to stock returns $\delta_{H,S}$ | 1.552 | 1.570 | 1.235 | 1.284 | 1.267 |
| | 0.000 * | 0.000 * | 0.000 * | 0.000 * | 0.000 * |
| Asymmetric spillover effect from HF price to stock returns $\theta_{H,S}$ | −0.083 | −0.122 | −0.099 | −0.039 | −0.087 |
| | 0.836 | 0.616 | 0.802 | 0.413 | 0.879 |
| Correlation coefficient $\rho$ | −0.268 | −0.186 | −0.241 | −0.141 | −0.160 |
| **Hong Kong** | **CFB-FE** | **CFB-T** | **CFB-HK** | **HFNV** | **HKP** |
| Volatility persistence stock returns $b_S$ | 0.947 | 0.946 | 0.947 | 0.947 | 0.947 |
| | 0.000 * | 0.000 * | 0.000 * | 0.000 * | 0.000 * |
| Spillover from stock returns to HF price $\delta_{S,H}$ | 0.068 | 0.057 | 0.065 | 0.080 | 0.057 |
| | 0.100 *** | 0.100 *** | 0.100 *** | 0.100 *** | 0.100 *** |
| Asymmetric spillover effect from stock returns to HF price $\theta_{S,H}$ | −0.022 | −0.010 | −0.020 | −0.012 | −0.008 |
| | 0.058 ** | 0.084 *** | 0.073 *** | 0.081 *** | 0.091 *** |
| Volatility persistence HF price $b_H$ | 0.473 | 0.517 | 0.593 | 0.538 | 0.507 |
| | 0.000 * | 0.000 * | 0.000 * | 0.000 * | 0.000 * |
| Spillover from HF price to stock returns $\delta_{H,S}$ | 1.217 | 1.480 | 1.611 | 1.515 | 1.404 |
| | 0.000 * | 0.000 * | 0.000 * | 0.000 * | 0.000 * |
| Asymmetric spillover effect from HF price to stock returns $\theta_{H,S}$ | −0.061 | −0.082 | −0.137 | −0.149 | −0.073 |
| | 0.556 | 0.820 | 0.330 | 0.305 | 0.751 |
| Correlation coefficient $\rho$ | −0.295 | −0.229 | −0.295 | −0.097 | −0.197 |
| **Hong Kong** | **IIF** | **IVI** | **IGF** | **JKAI** | **LIM** |
| Volatility persistence stock returns $b_S$ | 0.947 | 0.947 | 0.947 | 0.947 | 0.947 |
| | 0.000 * | 0.000 * | 0.000 * | 0.000 * | 0.000 * |
| Spillover from stock returns to HF price $\delta_{S,H}$ | 0.069 | 0.077 | 0.076 | 0.056 | 0.076 |
| | 0.100 *** | 0.100 *** | 0.100 *** | 0.100 *** | 0.100 *** |
| Asymmetric spillover effect from stock returns to HF price $\theta_{S,H}$ | −0.011 | −0.024 | −0.021 | 0.003 | −0.020 |
| | 0.084 *** | 0.040 ** | 0.054 ** | 0.092 *** | 0.071 *** |
| Volatility persistence HF price $b_H$ | 0.475 | 0.537 | 0.581 | 0.465 | 0.498 |
| | 0.000 * | 0.000 * | 0.000 * | 0.000 * | 0.000 * |
| Spillover from HF price to stock returns $\delta_{H,S}$ | 1.411 | 1.451 | 1.636 | 1.324 | 1.187 |
| | 0.000 * | 0.000 * | 0.000 * | 0.000 * | 0.000 * |
| Asymmetric spillover effect from HF price to stock returns $\theta_{H,S}$ | −0.057 | −0.064 | −0.064 | −0.082 | −0.102 |
| | 0.562 | 0.593 | 0.665 | 0.837 | 0.799 |
| Correlation coefficient $\rho$ | −0.174 | −0.164 | −0.235 | −0.142 | −0.283 |

**Table 9.** *Cont.*

| Hong Kong | MLM | PPL | PIF | PCF | ISF |
|---|---|---|---|---|---|
| Volatility persistence stock returns $b_S$ | 0.947 | 0.947 | 0.947 | 0.947 | 0.947 |
| | 0.000 * | 0.000 * | 0.000 * | 0.000 * | 0.000 * |
| Spillover from stock returns to HF price $\delta_{S,H}$ | 0.063 | 0.047 | 0.057 | 0.055 | 0.053 |
| | 0.100 *** | 0.100 *** | 0.100 *** | 0.100 *** | 0.100 *** |
| Asymmetric spillover effect from stock returns to HF price $\theta_{S,H}$ | 0.017 | 0.007 | 0.011 | 0.002 | 0.002 |
| | 0.088 *** | 0.093 *** | 0.092 *** | 0.092 *** | 0.092 *** |
| Volatility persistence HF price $b_H$ | 0.388 | 0.427 | 0.420 | 0.529 | 0.479 |
| | 0.000 * | 0.000 * | 0.000 * | 0.000 * | 0.000 * |
| Spillover from HF price to stock returns $\delta_{H,S}$ | 1.220 | 1.390 | 1.450 | 1.663 | 1.293 |
| | 0.000 * | 0.000 * | 0.000 * | 0.000 * | 0.000 * |
| Asymmetric spillover effect from HF price to stock returns $\theta_{H,S}$ | −0.117 | −0.127 | −0.062 | −0.081 | −0.091 |
| | 0.649 | 0.571 | 0.669 | 0.827 | 0.905 *** |
| Correlation coefficient $\rho$ | −0.182 | −0.276 | −0.239 | −0.205 | −0.139 |
| **Hong Kong** | **SJO** | **SRG** | **VPC-A** | **VPC-B** | **VEI** |
| Volatility persistence stock returns $b_S$ | 0.947 | 0.947 | 0.947 | 0.947 | 0.947 |
| | 0.000 * | 0.000 * | 0.000 * | 0.000 * | 0.000 * |
| Spillover from stock returns to HF price $\delta_{S,H}$ | 0.055 | 0.065 | 0.053 | 0.052 | 0.056 |
| | 0.100 *** | 0.100 *** | 0.100 *** | 0.100 *** | 0.100 *** |
| Asymmetric spillover effect from stock returns to HF price $\theta_{S,H}$ | 0.010 | 0.011 | 0.006 | 0.005 | 0.013 |
| | 0.091 *** | 0.090 *** | 0.092 *** | 0.092 *** | 0.089 *** |
| Volatility persistence HF price $b_H$ | 0.477 | 0.468 | 0.480 | 0.473 | 0.762 |
| | 0.000 * | 0.000 * | 0.000 * | 0.000 * | 0.000 * |
| Spillover from HF price to stock returns $\delta_{H,S}$ | 1.450 | 1.268 | 1.413 | 1.388 | 0.828 |
| | 0.000 * | 0.000 * | 0.000 * | 0.000 * | 0.000 * |
| Asymmetric spillover effect from HF price to stock returns $\theta_{H,S}$ | 0.154 | 0.112 | 0.130 | 0.131 | 0.100 |
| | 0.267 | 0.651 | 0.459 | 0.463 | 0.776 |
| Correlation coefficient $\rho$ | −0.154 | −0.162 | −0.241 | −0.243 | −0.091 |

* 1% significance level, ** 5% significance level, *** 10% significance level.

Table 10 shows the results of volatility analysis between the Thailand stock market and the hedge fund market and vice versa. For volatility persistence, the coefficient (b) is significant for the Thailand stock market. The volatility persistence for hedge fund market is also significant. In addition, the values of ($b_S$) and ($b_H$) are less than one, a condition necessary to have stable volatility, which suggests a long-term impact in both directions.

For volatility spillover, we find that the coefficient (δ) is significant, which suggests bidirectional volatility spillover exists between the Thailand stock market and the hedge fund market. For the asymmetric spillover response, the coefficient (θ) is insignificant for both the Thailand stock market on the hedge fund market and vice versa. This implies that negative shocks from the Thailand stock market do not impact the returns of hedge funds and neither does any negative news from hedge fund market affect the returns of the Thailand stock market.

**Table 10.** Volatility spillover between the Thailand Stock Market (SET) and Hedge Fund Returns.

| Thailand | AGF | AACF | AAGF | BDP | BAF |
|---|---|---|---|---|---|
| Volatility persistence stock returns $b_S$ | 0.748 | 0.748 | 0.748 | 0.747 | 0.746 |
| | 0.000 * | 0.000 * | 0.000 * | 0.000 * | 0.000 * |
| Spillover from stock returns to HF price $\delta_{S,H}$ | −0.145 | −0.145 | −0.142 | −0.138 | −0.132 |
| | 0.000 * | 0.000 * | 0.000 * | 0.000 * | 0.000 * |
| Asymmetric spillover effect from stock returns to HF price $\theta_{S,H}$ | −0.202 | −0.202 | −0.212 | −0.212 | −0.213 |
| | 0.291 | 0.292 | 0.290 | 0.290 | 0.289 |
| Volatility persistence HF price $b_H$ | 0.250 | 0.380 | 0.379 | 0.204 | 0.231 |
| | 0.000 * | 0.000 * | 0.000 * | 0.000 * | 0.000 * |
| Spillover from HF price to stock returns $\delta_{H,S}$ | 1.352 | 1.370 | 1.035 | 1.084 | 1.067 |
| | 0.000 * | 0.000 * | 0.000 * | 0.000 * | 0.000 * |
| Asymmetric spillover effect from HF price to stock returns $\theta_{H,S}$ | −0.283 | −0.322 | −0.299 | −0.239 | −0.287 |
| | 0.636 | 0.416 | 0.602 | 0.213 | 0.679 |
| Correlation coefficient $\rho$ | −0.468 | −0.386 | −0.441 | −0.341 | −0.360 |
| **Thailand** | **CFB-FE** | **CFB-T** | **CFB-HK** | **HFNV** | **HKP** |
| Volatility persistence stock returns $b_S$ | 0.747 | 0.746 | 0.747 | 0.747 | 0.747 |
| | 0.000 * | 0.000 * | 0.000 * | 0.000 * | 0.000 * |
| Spillover from stock returns to HF price $\delta_{S,H}$ | −0.132 | −0.143 | −0.135 | −0.120 | −0.143 |
| | 0.000 * | 0.000 * | 0.000 * | 0.000 * | 0.000 * |
| Asymmetric spillover effect from stock returns to HF price $\theta_{S,H}$ | −0.222 | −0.210 | −0.220 | −0.212 | −0.208 |
| | 0.258 | 0.284 | 0.273 | 0.281 | 0.291 |
| Volatility persistence HF price $b_H$ | 0.273 | 0.317 | 0.393 | 0.338 | 0.307 |
| | 0.000 * | 0.000 * | 0.000 * | 0.000 * | 0.000 * |
| Spillover from HF price to stock returns $\delta_{H,S}$ | 1.017 | 1.280 | 1.411 | 1.315 | 1.204 |
| | 0.000 * | 0.000 * | 0.000 * | 0.000 * | 0.000 * |
| Asymmetric spillover effect from HF price to stock returns $\theta_{H,S}$ | −0.261 | −0.282 | −0.337 | −0.349 | −0.273 |
| | 0.356 | 0.620 | 0.130 | 0.105 | 0.551 |
| Correlation coefficient $\rho$ | −0.495 | −0.429 | −0.495 | −0.297 | −0.397 |
| **Thailand** | **IIF** | **IVI** | **IGF** | **JKAI** | **LIM** |
| Volatility persistence stock returns $b_S$ | 0.747 | 0.747 | 0.747 | 0.747 | 0.747 |
| | 0.000 * | 0.000 * | 0.000 * | 0.000 * | 0.000 * |
| Spillover from stock returns to HF price $\delta_{S,H}$ | −0.131 | −0.123 | −0.124 | −0.144 | −0.124 |
| | 0.000 * | 0.000 * | 0.000 * | 0.000 * | 0.000 * |
| Asymmetric spillover effect from stock returns to HF price $\theta_{S,H}$ | −0.211 | −0.224 | −0.221 | −0.197 | −0.220 |
| | 0.284 | 0.240 | 0.254 | 0.292 | 0.271 |
| Volatility persistence HF price $b_H$ | 0.275 | 0.337 | 0.381 | 0.265 | 0.298 |
| | 0.000 * | 0.000 * | 0.000 * | 0.000 * | 0.000 * |
| Spillover from HF price to stock returns $\delta_{H,S}$ | 1.211 | 1.251 | 1.436 | 1.124 | 0.987 |
| | 0.000 * | 0.000 * | 0.000 * | 0.000 * | 0.000 * |
| Asymmetric spillover effect from HF price to stock returns $\theta_{H,S}$ | −0.257 | −0.264 | −0.264 | −0.282 | −0.302 |
| | 0.362 | 0.393 | 0.465 | 0.637 | 0.599 |
| Correlation coefficient $\rho$ | −0.374 | −0.364 | −0.435 | −0.342 | −0.483 |

**Table 10.** *Cont.*

| Thailand | MLM | PPL | PIF | PCF | ISF |
|---|---|---|---|---|---|
| Volatility persistence stock returns $b_S$ | 0.747 | 0.747 | 0.747 | 0.747 | 0.747 |
| | 0.000 * | 0.000 * | 0.000 * | 0.000 * | 0.000 * |
| Spillover from stock returns to HF price $\delta_{S,H}$ | −0.137 | −0.153 | −0.143 | −0.145 | −0.147 |
| | 0.000 * | 0.000 * | 0.000 * | 0.000 * | 0.000 * |
| Asymmetric spillover effect from stock returns to HF price $\theta_{S,H}$ | −0.217 | −0.193 | −0.211 | −0.198 | −0.198 |
| | 0.288 | 0.293 | 0.292 | 0.292 | 0.292 |
| Volatility persistence HF price $b_H$ | 0.188 | 0.227 | 0.220 | 0.329 | 0.279 |
| | 0.000 * | 0.000 * | 0.000 * | 0.000 * | 0.000 * |
| Spillover from HF price to stock returns $\delta_{H,S}$ | 1.020 | 1.190 | 1.250 | 1.463 | 1.093 |
| | 0.000 * | 0.000 * | 0.000 * | 0.000 * | 0.000 * |
| Asymmetric spillover effect from HF price to stock returns $\theta_{H,S}$ | −0.317 | −0.327 | −0.262 | −0.281 | −0.291 |
| | 0.449 | 0.371 | 0.469 | 0.627 | 0.705 |
| Correlation coefficient $\rho$ | −0.382 | −0.476 | −0.439 | −0.405 | −0.339 |
| **Thailand** | **SJO** | **SRG** | **VPC-A** | **VPC-B** | **VEI** |
| Volatility persistence stock returns $b_S$ | 0.747 | 0.747 | 0.747 | 0.747 | 0.747 |
| | 0.000 * | 0.000 * | 0.000 * | 0.000 * | 0.000 * |
| Spillover from stock returns to HF price $\delta_{S,H}$ | −0.145 | −0.135 | −0.147 | −0.148 | −0.144 |
| | 0.000 * | 0.000 * | 0.000 * | 0.000 * | 0.000 * |
| Asymmetric spillover effect from stock returns to HF price $\theta_{S,H}$ | −0.210 | −0.211 | −0.206 | −0.205 | −0.213 |
| | 0.291 | 0.290 | 0.292 | 0.292 | 0.289 |
| Volatility persistence HF price $b_H$ | 0.277 | 0.268 | 0.280 | 0.273 | 0.562 |
| | 0.000 * | 0.000 * | 0.000 * | 0.000 * | 0.000 * |
| Spillover from HF price to stock returns $\delta_{H,S}$ | 1.250 | 1.068 | 1.213 | 1.188 | 0.628 |
| | 0.000 * | 0.000 * | 0.000 * | 0.000 * | 0.000 * |
| Asymmetric spillover effect from HF price to stock returns $\theta_{H,S}$ | −0.354 | −0.312 | −0.330 | −0.331 | −0.300 |
| | 0.067 *** | 0.451 | 0.259 | 0.263 | 0.576 |
| Correlation coefficient $\rho$ | −0.354 | −0.362 | −0.441 | −0.443 | −0.291 |

\* 1% significance level, \*\*\* 10% significance level.

Table 11 shows the results of the volatility analysis between the Indonesian stock market and the hedge fund market and vice versa. For volatility persistence, the coefficient (b) is significant for the Indonesian stock market. The volatility persistence for hedge funds is also significant. In addition, the values of ($b_S$) and ($b_H$) are less than one, a condition necessarily have stable volatility with a long-term impact on market returns.

For volatility spillover, we find that the coefficient ($\delta$) is significant for the Indonesian stock market on the hedge fund market and vice versa at 1% level of significance. This implies the return spillover effect is strong in both directions from the hedge fund market to the Indonesian stock market and from the Indonesian stock market to hedge fund market. This knowledge of volatility spillover effects can be helpful in asset allocation and stock selection.

For the asymmetric spillover response, the coefficient ($\theta$) is insignificant for the Indonesian stock market on the hedge fund market and the reciprocal is also insignificant. This implies that negative shocks from the Indonesian stock market do not impact volatility in the hedge fund market and no reciprocal volatility impact can be seen between the hedge fund market and the Indonesia stock market.

**Table 11.** Volatility spillover between the Indonesian Stock Market (JCI) and Hedge Fund Returns.

| Indonesia | AGF | AACF | AAGF | BDP | BAF |
|---|---|---|---|---|---|
| Volatility persistence stock returns $b_S$ | 0.753 | 0.753 | 0.753 | 0.752 | 0.751 |
| | 0.000 * | 0.000 * | 0.000 * | 0.000 * | 0.000 * |
| Spillover from stock returns to HF price $\delta_{S,H}$ | −0.140 | −0.140 | −0.137 | −0.133 | −0.127 |
| | 0.000 * | 0.000 * | 0.000 * | 0.000 * | 0.000 * |
| Asymmetric spillover effect from stock returns to HF price $\theta_{S,H}$ | −0.197 | −0.198 | −0.207 | −0.207 | −0.208 |
| | 0.286 | 0.287 | 0.285 | 0.285 | 0.284 |
| Volatility persistence HF price $b_H$ | 0.255 | 0.385 | 0.384 | 0.209 | 0.236 |
| | 0.000 * | 0.000 * | 0.000 * | 0.000 * | 0.000 * |
| Spillover from HF price to stock returns $\delta_{H,S}$ | 1.357 | 1.375 | 1.040 | 1.089 | 1.072 |
| | 0.000 * | 0.000 * | 0.000 * | 0.000 * | 0.000 * |
| Asymmetric spillover effect from HF price to stock returns $\theta_{H,S}$ | −0.278 | −0.317 | −0.294 | −0.234 | −0.282 |
| | 0.641 | 0.420 | 0.606 | 0.218 | 0.684 |
| Correlation coefficient ρ | −0.463 | −0.381 | −0.436 | −0.336 | −0.355 |
| **Indonesia** | **CFB-FE** | **CFB-T** | **CFB-HK** | **HFNV** | **HKP** |
| Volatility persistence stock returns $b_S$ | 0.752 | 0.751 | 0.752 | 0.752 | 0.752 |
| | 0.000 * | 0.000 * | 0.000 * | 0.000 * | 0.000 * |
| Spillover from stock returns to HF price $\delta_{S,H}$ | −0.127 | −0.138 | −0.130 | −0.115 | −0.138 |
| | 0.000 * | 0.000 * | 0.000 * | 0.000 * | 0.000 * |
| Asymmetric spillover effect from stock returns to HF price $\theta_{S,H}$ | −0.217 | −0.205 | −0.215 | −0.207 | −0.204 |
| | 0.253 | 0.280 | 0.268 | 0.276 | 0.287 |
| Volatility persistence HF price $b_H$ | 0.278 | 0.322 | 0.397 | 0.343 | 0.312 |
| | 0.000 * | 0.000 * | 0.000 * | 0.000 * | 0.000 * |
| Spillover from HF price to stock returns $\delta_{H,S}$ | 1.022 | 1.285 | 1.416 | 1.320 | 1.209 |
| | 0.000 * | 0.000 * | 0.000 * | 0.000 * | 0.000 * |
| Asymmetric spillover effect from HF price to stock returns $\theta_{H,S}$ | −0.256 | −0.277 | −0.332 | −0.344 | −0.268 |
| | 0.361 | 0.625 | 0.135 | 0.110 | 0.556 |
| Correlation coefficient ρ | −0.490 | −0.424 | −0.490 | −0.292 | −0.392 |
| **Indonesia** | **IIF** | **IVI** | **IGF** | **JKAI** | **LIM** |
| Volatility persistence stock returns $b_S$ | 0.752 | 0.752 | 0.752 | 0.752 | 0.752 |
| | 0.000 * | 0.000 * | 0.000 * | 0.000 * | 0.000 * |
| Spillover from stock returns to HF price $\delta_{S,H}$ | −0.127 | −0.118 | −0.119 | −0.139 | −0.119 |
| | 0.000 * | 0.000 * | 0.000 * | 0.000 * | 0.000 * |
| Asymmetric spillover effect from stock returns to HF price $\theta_{S,H}$ | −0.206 | −0.219 | −0.216 | −0.193 | −0.215 |
| | 0.279 | 0.235 | 0.250 | 0.287 | 0.266 |
| Volatility persistence HF price $b_H$ | 0.280 | 0.342 | 0.386 | 0.270 | 0.303 |
| | 0.000 * | 0.000 * | 0.000 * | 0.000 * | 0.000 * |
| Spillover from HF price to stock returns $\delta_{H,S}$ | 1.216 | 1.256 | 1.441 | 1.129 | 0.992 |
| | 0.000 * | 0.000 * | 0.000 * | 0.000 * | 0.000 * |
| Asymmetric spillover effect from HF price to stock returns $\theta_{H,S}$ | −0.252 | −0.259 | −0.259 | −0.277 | −0.297 |
| | 0.366 | 0.398 | 0.470 | 0.642 | 0.604 |
| Correlation coefficient ρ | −0.369 | −0.359 | −0.430 | −0.337 | −0.478 |

**Table 11.** *Cont.*

| Indonesia | MLM | PPL | PIF | PCF | ISF |
|---|---|---|---|---|---|
| Volatility persistence stock returns $b_S$ | 0.752 | 0.752 | 0.752 | 0.752 | 0.752 |
| | 0.000 * | 0.000 * | 0.000 * | 0.000 * | 0.000 * |
| Spillover from stock returns to HF price $\delta_{S,H}$ | −0.132 | −0.148 | −0.138 | −0.140 | −0.142 |
| | 0.000 * | 0.000 * | 0.000 * | 0.000 * | 0.000 * |
| Asymmetric spillover effect from stock returns to HF price $\theta_{S,H}$ | −0.212 | −0.188 | −0.206 | −0.193 | −0.193 |
| | 0.283 | 0.288 | 0.287 | 0.287 | 0.287 |
| Volatility persistence HF price $b_H$ | 0.193 | 0.232 | 0.225 | 0.334 | 0.284 |
| | 0.000 * | 0.000 * | 0.000 * | 0.000 * | 0.000 * |
| Spillover from HF price to stock returns $\delta_{H,S}$ | 1.024 | 1.195 | 1.255 | 1.468 | 1.097 |
| | 0.000 * | 0.000 * | 0.000 * | 0.000 * | 0.000 * |
| Asymmetric spillover effect from HF price to stock returns $\theta_{H,S}$ | −0.312 | −0.322 | −0.257 | −0.276 | −0.286 |
| | 0.454 | 0.376 | 0.473 | 0.632 | 0.709 |
| Correlation coefficient $\rho$ | −0.378 | −0.471 | −0.434 | −0.400 | −0.334 |
| **Indonesia** | **SJO** | **SRG** | **VPC-A** | **VPC-B** | **VEI** |
| Volatility persistence stock returns $b_S$ | 0.752 | 0.752 | 0.752 | 0.752 | 0.752 |
| | 0.000 * | 0.000 * | 0.000 * | 0.000 * | 0.000 * |
| Spillover from stock returns to HF price $\delta_{S,H}$ | −0.140 | −0.130 | −0.142 | −0.143 | −0.139 |
| | 0.000 * | 0.000 * | 0.000 * | 0.000 * | 0.000 * |
| Asymmetric spillover effect from stock returns to HF price $\theta_{S,H}$ | −0.205 | −0.206 | −0.201 | −0.200 | −0.208 |
| | 0.286 | 0.285 | 0.287 | 0.287 | 0.284 |
| Volatility persistence HF price $b_H$ | 0.282 | 0.273 | 0.285 | 0.278 | 0.567 |
| | 0.000 * | 0.000 * | 0.000 * | 0.000 * | 0.000 * |
| Spillover from HF price to stock returns $\delta_{H,S}$ | 1.255 | 1.073 | 1.217 | 1.193 | 0.633 |
| | 0.000 * | 0.000 * | 0.000 * | 0.000 * | 0.000 * |
| Asymmetric spillover effect from HF price to stock returns $\theta_{H,S}$ | −0.349 | −0.307 | −0.326 | −0.326 | −0.295 |
| | 0.072 *** | 0.455 | 0.264 | 0.267 | 0.581 |
| Correlation coefficient $\rho$ | −0.349 | −0.357 | −0.436 | −0.438 | −0.286 |

* 1% significance level, *** 10% significance level.

Table 12 shows the results of volatility analysis between the Malaysian stock market and the hedge fund market and vice versa. For volatility persistence, the coefficient (b) is significant for the Malaysian stock market. Volatility persistence for the hedge fund market is also significant at the 1% level.

For volatility spillover, we find that the coefficient (δ) is significant for the Malaysian stock market on hedge fund market and for the hedge fund market on the Malaysian stock market the spillover effect was significant at the 1% level. This implies the return spillover effect is strong from the hedge fund market to the Malaysian stock market and vice versa.

For the asymmetric spillover response, the coefficient (θ) is insignificant for the Malaysian stock market on the hedge fund market and the reciprocal is insignificant. This implies that negative shocks generated from negative news in the Malaysian stock market do not impact the volatility of the hedge fund market and vice versa.

**Table 12.** Volatility spillover between the Malaysian Stock Market (FBMKLCI) and Hedge Fund Returns.

| Malaysia | AGF | AACF | AAGF | BDP | BAF |
|---|---|---|---|---|---|
| Volatility persistence stock returns $b_S$ | 0.843 | 0.843 | 0.843 | 0.842 | 0.841 |
| | 0.000 * | 0.000 * | 0.000 * | 0.000 * | 0.000 * |
| Spillover from stock returns to HF price $\delta_{S,H}$ | −0.050 | −0.050 | −0.047 | −0.043 | −0.037 |
| | 0.000 * | 0.000 * | 0.000 * | 0.000 * | 0.000 * |
| Asymmetric spillover effect from stock returns to HF price $\theta_{S,H}$ | −0.107 | −0.108 | −0.117 | −0.117 | −0.118 |
| | 0.196 | 0.197 | 0.195 | 0.195 | 0.194 |
| Volatility persistence HF price $b_H$ | 0.345 | 0.475 | 0.474 | 0.299 | 0.326 |
| | 0.000 * | 0.000 * | 0.000 * | 0.000 * | 0.000 * |
| Spillover from HF price to stock returns $\delta_{H,S}$ | 1.447 | 1.465 | 1.130 | 1.179 | 1.162 |
| | 0.000 * | 0.000 * | 0.000 * | 0.000 * | 0.000 * |
| Asymmetric spillover effect from HF price to stock returns $\theta_{H,S}$ | −0.188 | −0.227 | −0.204 | −0.144 | −0.192 |
| | 0.731 | 0.510 | 0.696 | 0.308 | 0.774 |
| Correlation coefficient ρ | −0.373 | −0.291 | −0.346 | −0.246 | −0.265 |
| **Malaysia** | **CFB-FE** | **CFB-T** | **CFB-HK** | **HFNV** | **HKP** |
| Volatility persistence stock returns $b_S$ | 0.842 | 0.841 | 0.842 | 0.842 | 0.842 |
| | 0.000 * | 0.000 * | 0.000 * | 0.000 * | 0.000 * |
| Spillover from stock returns to HF price $\delta_{S,H}$ | −0.037 | −0.048 | −0.040 | −0.025 | −0.048 |
| | 0.000 * | 0.000 * | 0.000 * | 0.000 * | 0.000 * |
| Asymmetric spillover effect from stock returns to HF price $\theta_{S,H}$ | −0.127 | −0.115 | −0.125 | −0.117 | −0.114 |
| | 0.163 | 0.190 | 0.178 | 0.186 | 0.197 |
| Volatility persistence HF price $b_H$ | 0.368 | 0.412 | 0.487 | 0.433 | 0.402 |
| | 0.000 * | 0.000 * | 0.000 * | 0.000 * | 0.000 * |
| Spillover from HF price to stock returns $\delta_{H,S}$ | 1.112 | 1.375 | 1.506 | 1.410 | 1.299 |
| | 0.000 * | 0.000 * | 0.000 * | 0.000 * | 0.000 * |
| Asymmetric spillover effect from HF price to stock returns $\theta_{H,S}$ | −0.166 | −0.187 | −0.242 | −0.254 | −0.178 |
| | 0.451 | 0.715 | 0.225 | 0.200 | 0.646 |
| Correlation coefficient ρ | −0.400 | −0.334 | −0.400 | −0.202 | −0.302 |
| **Malaysia** | **IIF** | **IVI** | **IGF** | **JKAI** | **LIM** |
| Volatility persistence stock returns $b_S$ | 0.842 | 0.842 | 0.842 | 0.842 | 0.842 |
| | 0.000 * | 0.000 * | 0.000 * | 0.000 * | 0.000 * |
| Spillover from stock returns to HF price $\delta_{S,H}$ | −0.037 | −0.028 | −0.029 | −0.049 | −0.029 |
| | 0.000 * | 0.000 * | 0.000 * | 0.000 * | 0.000 * |
| Asymmetric spillover effect from stock returns to HF price $\theta_{S,H}$ | −0.116 | −0.129 | −0.126 | −0.103 | −0.125 |
| | 0.189 | 0.145 | 0.160 | 0.197 | 0.176 |
| Volatility persistence HF price $b_H$ | 0.370 | 0.432 | 0.476 | 0.360 | 0.393 |
| | 0.000 | 0.000 * | 0.000 * | 0.000 * | 0.000 * |
| Spillover from HF price to stock returns $\delta_{H,S}$ | 1.306 | 1.346 | 1.531 | 1.219 | 1.082 |
| | 0.000 * | 0.000 * | 0.000 * | 0.000 * | 0.000 * |
| Asymmetric spillover effect from HF price to stock returns $\theta_{H,S}$ | −0.162 | −0.169 | −0.169 | −0.187 | −0.207 |
| | 0.456 | 0.488 | 0.560 | 0.732 | 0.694 |
| Correlation coefficient ρ | −0.279 | −0.269 | −0.340 | −0.247 | −0.388 |

**Table 12.** *Cont.*

| Malaysia | MLM | PPL | PIF | PCF | ISF |
|---|---|---|---|---|---|
| Volatility persistence stock returns $b_S$ | 0.842 | 0.842 | 0.842 | 0.842 | 0.842 |
| | 0.000 * | 0.000 * | 0.000 * | 0.000 * | 0.000 * |
| Spillover from stock returns to HF price $\delta_{S,H}$ | −0.042 | −0.058 | −0.048 | −0.050 | −0.052 |
| | 0.000 * | 0.000 * | 0.000 * | 0.000 * | 0.000 * |
| Asymmetric spillover effect from stock returns to HF price $\theta_{S,H}$ | −0.122 | −0.098 | −0.116 | −0.103 | −0.103 |
| | 0.193 | 0.198 | 0.197 | 0.197 | 0.1970 |
| Volatility persistence HF price $b_H$ | 0.283 | 0.322 | 0.315 | 0.424 | 0.374 |
| | 0.000 * | 0.000 * | 0.000 * | 0.000 * | 0.000 * |
| Spillover from HF price to stock returns $\delta_{H,S}$ | 1.114 | 1.285 | 1.345 | 1.558 | 1.187 |
| | 0.000 * | 0.000 * | 0.000 * | 0.000 * | 0.000 * |
| Asymmetric spillover effect from HF price to stock returns $\theta_{H,S}$ | −0.222 | −0.232 | −0.167 | −0.186 | −0.196 |
| | 0.544 | 0.466 | 0.563 | 0.722 | 0.799 |
| Correlation coefficient $\rho$ | −0.288 | −0.381 | −0.344 | −0.310 | −0.244 |
| **Malaysia** | **SJO** | **SRG** | **VPC-A** | **VPC-B** | **VEI** |
| Volatility persistence stock returns $b_S$ | 0.842 | 0.842 | 0.842 | 0.842 | 0.842 |
| | 0.000 * | 0.000 * | 0.000 * | 0.000 * | 0.000 * |
| Spillover from stock returns to HF price $\delta_{S,H}$ | −0.050 | −0.040 | −0.052 | −0.053 | −0.049 |
| | 0.000 * | 0.000 * | 0.000 * | 0.000 * | 0.000 * |
| Asymmetric spillover effect from stock returns to HF price $\theta_{S,H}$ | −0.115 | −0.116 | −0.111 | −0.110 | −0.118 |
| | 0.196 | 0.195 | 0.197 | 0.197 | 0.194 |
| Volatility persistence HF price $b_H$ | 0.372 | 0.363 | 0.375 | 0.368 | 0.657 |
| | 0.000 * | 0.000 * | 0.000 * | 0.000 * | 0.000 * |
| Spillover from HF price to stock returns $\delta_{H,S}$ | 1.345 | 1.163 | 1.307 | 1.283 | 0.723 |
| | 0.000 * | 0.000 * | 0.000 * | 0.000 * | 0.0000 * |
| Asymmetric spillover effect from HF price to stock returns $\theta_{H,S}$ | −0.259 | −0.217 | −0.236 | −0.236 | −0.205 |
| | 0.162 | 0.545 | 0.354 | 0.357 | 0.671 |
| Correlation coefficient $\rho$ | −0.259 | −0.267 | −0.346 | −0.348 | −0.196 |

* 1% significance level.

Table 13 shows the results of the volatility analysis between the Taiwanese stock market and the hedge fund market and vice versa. For volatility persistence, the coefficient (b) is significant for the Taiwanese stock market. The volatility persistence for hedge fund market is also significant at the 1% level.

For volatility spillover, we find that the coefficient ($\delta$) is significant for the Taiwanese stock market on hedge fund market and the hedge fund market on the Taiwanese stock market shows a spillover effect at 1% significance level. For the asymmetric spillover response, the coefficient ($\theta$) is insignificant for the Taiwanese stock market on the hedge fund market, with the exception of IVI and IGF where the asymmetric spillover coefficient is significant at the 10% level. This implies that negative shocks from the Taiwanese stock market do not impact the returns of hedge funds and neither does any negative news from hedge fund market affect the returns of the Taiwanese stock market except for two of 25 studied hedge funds that can be taken as a weak impact.

**Table 13.** Volatility spillover between the Taiwanese Stock Market (TWSE) and Hedge Fund Returns.

| Taiwan | AGF | AACF | AAGF | BDP | BAF |
|---|---|---|---|---|---|
| Volatility persistence stock returns $b_S$ | 0.903 | 0.903 | 0.903 | 0.902 | 0.901 |
| | 0.000 * | 0.000 * | 0.000 * | 0.000 * | 0.000 * |
| Spillover from stock returns to HF price $\delta_{S,H}$ | 0.010 | 0.010 | 0.013 | 0.017 | 0.023 |
| | 0.000 * | 0.000 * | 0.000 * | 0.000 * | 0.000 * |
| Asymmetric spillover effect from stock returns to HF price $\theta_{S,H}$ | −0.047 | −0.048 | −0.057 | −0.057 | −0.058 |
| | 0.136 | 0.137 | 0.135 | 0.135 | 0.134 |
| Volatility persistence HF price $b_H$ | 0.405 | 0.535 | 0.534 | 0.359 | 0.386 |
| | 0.000 * | 0.000 * | 0.000 * | 0.000 * | 0.000 * |
| Spillover from HF price to stock returns $\delta_{H,S}$ | 1.507 | 1.525 | 1.190 | 1.239 | 1.222 |
| | 0.000 * | 0.000 * | 0.000 * | 0.000 * | 0.000 * |
| Asymmetric spillover effect from HF price to stock returns $\theta_{H,S}$ | −0.128 | −0.167 | −0.144 | −0.084 | −0.132 |
| | 0.791 | 0.570 | 0.756 | 0.368 | 0.834 |
| Correlation coefficient $\rho$ | −0.313 | −0.231 | −0.286 | −0.186 | −0.205 |
| **Taiwan** | **CFB-FE** | **CFB-T** | **CFB-HK** | **HFNV** | **HKP** |
| Volatility persistence stock returns $b_S$ | 0.902 | 0.901 | 0.902 | 0.902 | 0.902 |
| | 0.000 * | 0.000 * | 0.000 * | 0.000 * | 0.000 * |
| Spillover from stock returns to HF price $\delta_{S,H}$ | 0.023 | 0.012 | 0.020 | 0.035 | 0.012 |
| | 0.000 * | 0.0000 * | 0.000 * | 0.000 * | 0.000 * |
| Asymmetric spillover effect from stock returns to HF price $\theta_{S,H}$ | −0.067 | −0.055 | −0.065 | −0.057 | −0.054 |
| | 0.103 | 0.130 | 0.118 | 0.126 | 0.137 |
| Volatility persistence HF price $b_H$ | 0.428 | 0.472 | 0.547 | 0.493 | 0.462 |
| | 0.000 * | 0.000 * | 0.000 * | 0.000 * | 0.000 * |
| Spillover from HF price to stock returns $\delta_{H,S}$ | 1.172 | 1.435 | 1.566 | 1.470 | 1.359 |
| | 0.000 * | 0.000 * | 0.000 * | 0.000 * | 0.000 * |
| Asymmetric spillover effect from HF price to stock returns $\theta_{H,S}$ | −0.106 | −0.127 | −0.182 | −0.194 | −0.118 |
| | 0.511 | 0.775 | 0.285 | 0.260 | 0.706 |
| Correlation coefficient $\rho$ | −0.340 | −0.274 | −0.340 | −0.142 | −0.242 |
| **Taiwan** | **IIF** | **IVI** | **IGF** | **JKAI** | **LIM** |
| Volatility persistence stock returns $b_S$ | 0.902 | 0.902 | 0.902 | 0.902 | 0.902 |
| | 0.000 * | 0.000 * | 0.000 * | 0.000 * | 0.0000 * |
| Spillover from stock returns to HF price $\delta_{S,H}$ | 0.023 | 0.032 | 0.031 | 0.011 | 0.031 |
| | 0.000 * | 0.000 * | 0.000 * | 0.000 | 0.000 * |
| Asymmetric spillover effect from stock returns to HF price $\theta_{S,H}$ | −0.056 | −0.069 | −0.066 | −0.043 | −0.065 |
| | 0.129 | 0.085 *** | 0.100 *** | 0.137 | 0.116 |
| Volatility persistence HF price $b_H$ | 0.430 | 0.492 | 0.536 | 0.420 | 0.453 |
| | 0.000 * | 0.000 * | 0.000 * | 0.000 * | 0.000 * |
| Spillover from HF price to stock returns $\delta_{H,S}$ | 1.366 | 1.406 | 1.591 | 1.279 | 1.142 |
| | 0.000 * | 0.000 * | 0.000 * | 0.000 * | 0.000 * |
| Asymmetric spillover effect from HF price to stock returns $\theta_{H,S}$ | −0.102 | −0.109 | −0.109 | −0.127 | −0.147 |
| | 0.516 | 0.548 | 0.620 | 0.792 | 0.754 |
| Correlation coefficient $\rho$ | −0.219 | −0.209 | −0.280 | −0.187 | −0.328 |

**Table 13.** *Cont.*

| Taiwan | MLM | PPL | PIF | PCF | ISF |
|---|---|---|---|---|---|
| Volatility persistence stock returns $b_S$ | 0.902 | 0.902 | 0.902 | 0.902 | 0.902 |
| | 0.000 * | 0.000 * | 0.000 * | 0.000 * | 0.000 * |
| Spillover from stock returns to HF price $\delta_{S,H}$ | 0.018 | 0.002 | 0.012 | 0.010 | 0.008 |
| | 0.000 * | 0.000 * | 0.000 * | 0.000 * | 0.000 * |
| Asymmetric spillover effect from stock returns to HF price $\theta_{S,H}$ | −0.062 | −0.038 | −0.056 | −0.043 | −0.043 |
| | 0.133 | 0.138 | 0.137 | 0.137 | 0.137 |
| Volatility persistence HF price $b_H$ | 0.343 | 0.382 | 0.375 | 0.484 | 0.434 |
| | 0.000 * | 0.000 * | 0.000 * | 0.000 * | 0.000 * |
| Spillover from HF price to stock returns $\delta_{H,S}$ | 1.174 | 1.345 | 1.405 | 1.618 | 1.247 |
| | 0.000 * | 0.000 * | 0.000 * | 0.000 * | 0.000 * |
| Asymmetric spillover effect from HF price to stock returns $\theta_{H,S}$ | −0.162 | −0.172 | −0.107 | −0.126 | −0.136 |
| | 0.604 | 0.526 | 0.623 | 0.782 | 0.859 |
| Correlation coefficient $\rho$ | −0.228 | −0.321 | −0.284 | −0.250 | −0.184 |
| **Taiwan** | **SJO** | **SRG** | **VPC-A** | **VPC-B** | **VEI** |
| Volatility persistence stock returns $b_S$ | 0.902 | 0.902 | 0.902 | 0.902 | 0.902 |
| | 0.000 * | 0.000 * | 0.000 * | 0.000 * | 0.000 * |
| Spillover from stock returns to HF price $\delta_{S,H}$ | 0.010 | 0.020 | 0.008 | 0.007 | 0.011 |
| | 0.000 * | 0.000 * | 0.000 * | 0.000 * | 0.000 * |
| Asymmetric spillover effect from stock returns to HF price $\theta_{S,H}$ | −0.055 | −0.056 | −0.051 | −0.050 | −0.058 |
| | 0.136 | 0.135 | 0.137 | 0.137 | 0.134 |
| Volatility persistence HF price $b_H$ | 0.432 | 0.423 | 0.435 | 0.428 | 0.717 |
| | 0.000 * | 0.000 * | 0.000 * | 0.000 * | 0.000 * |
| Spillover from HF price to stock returns $\delta_{H,S}$ | 1.405 | 1.223 | 1.367 | 1.343 | 0.783 |
| | 0.000 * | 0.000 * | 0.000 * | 0.000 * | 0.000 * |
| Asymmetric spillover effect from HF price to stock returns $\theta_{H,S}$ | −0.199 | −0.157 | −0.176 | −0.176 | −0.145 |
| | 0.222 | 0.605 | 0.414 | 0.417 | 0.731 |
| Correlation coefficient $\rho$ | −0.199 | −0.207 | −0.286 | −0.288 | −0.136 |

* 1% significance level, *** 10% significance level.

### 4.4. Volatility Spillover

The volatility spillover results for the stock price and hedge fund coefficients are presented in Tables 2–13. The results of volatility spillover suggests that hedge funds are stable and independent regardless of stock market shocks in terms of changing their market position. These results are not surprising and are consistent with a previous study by Sung et al. (2021), who found that hedge fund exit financial markets simultaneously after financial stability shocks occur. This reiterates the concept of active fund management. Hedge funds use different investment strategies to invest for different time periods long- and short-term positions). As fund managers are paid based on performance incentives, they are more likely to adapt and adjust their strategies based on market conditions.

### 4.5. Asymmetric Spillover

The asymmetric spillover results for the stock and hedge fund market prices are presented in Tables 2–13. Overall, the results indicate that the asymmetric spillover coefficients from stock prices to hedge funds are significant for all sampled countries. However, the

asymmetric spillover coefficients from the hedge funds to the stock market are insignificant. This indicates that good news does not impact performance or spillover from the stock market to the hedge fund market. As hedge funds are managed by fund managers who use multiple investment strategies, we expected to see adjustments in hedge fund portfolios with good or bad news spread in financial markets because of active management and the benchmark of the hedge fund performance, fund managers need to meet. Hedge funds are based on risk-adjusted performance fees and incentive fees for fund managers, which is another motivation for changing investment strategies as per market conditions. This can is confirmed by the study by Ackermann et al. (1999), who conducted the research to identify managerial compensation and fee structure. They found strong evidence on managerial ability to take advantage of market liquidity and concluded there was persistence in timing skill over time.

The insignificant coefficients confirm that the spillover effect is symmetric, which suggests that positive and negative shocks have a similar impact on volatility. A decrease in stock market returns has a similar impact on hedge fund volatility as in increasing stock returns. The asymmetric spillover test also helps to identify the impact of news on the returns' volatility. For this study, we found negative signs against the significant asymmetric coefficients between the Chinese stock market and hedge fund market and between the Hong Kong stock market and hedge fund market, which indicates that good news on the stock market has a positive impact on other markets, which in this case, is the hedge fund market. For volatility spillover effects from stock returns to hedge funds, we did not find any strong evidence of stock market shocks driving hedge fund volatility. This suggests that there is no integration between the two markets. The weak or even negative correlations of the hedge fund market with stock markets allows the diversification of risk in a mean-variance environment.

The results do not show any significant evidence of asymmetric spillover effects for the sample countries in our study. We also did not find that good news related to stock prices had a significant impact on hedge funds. Nor did we find evidence of the reciprocal effect. The results also revealed a lack of volatility spillover between stock markets and hedge fund markets; investors should consider diversifying their investment portfolios by investing in the hedge fund market. However, it is important to understand hedge fund managers' different strategies before investing in this market. Investors must also be aware that these funds are actively managed funds so investors need to be more qualified to manage investments in hedge funds or use a hedge fund manager, but this can be expensive. The study by Philippon (2012) showed trading costs have decreased but the costs of active fund management are large. French (2008) estimated that investors spend 0.67% of asset value trying in vain, by definition to beat the market.

## 5. Conclusions

Relationships between stock market and hedge funds are of particular interest for academics and practitioners due to the fact that these two variables play a crucial role in portfolio and risk management. This paper also examined the volatility linkages between the stock and hedge fund markets. The findings show insignificant coefficients for volatility spillovers for all the hedge funds included in the study. We analysed twelve countries in the Asia-Pacific region, and it is worth mentioning that our overall conclusion, based on the literature, tended to be consistent with the absence of a relationship between stock market returns and hedge fund returns. For example, Martin (2006) studied the correlation between US hedge funds, S&P 500, and MSCI World Index and found a weak correlation between hedge funds, stocks and bonds. The author concluded that because of the low correlation, the integration of hedge funds into portfolios of traditional investments seems promising. Stulz (2007) concluded that hedge funds appear to be an attractive diversification vehicle for investors who hold stocks. Balakrishna (2012) found evidence to conclude that hedge funds provide significant diversification benefits, since these funds have low correlations with conventional asset classes over the business cycle. It is also important to understand

that the hedge funds' behaviour is considered independent and influential simultaneously. Specifically, during the 1997 Asian financial crisis, numerous researchers studied whether hedge funds caused or contributed to the financial instability or the market crash e.g., Park et al. (1998); Brown et al. (2000). Brown et al. (2000) investigated the 1997 Asian financial crisis and hedge funds. The authors found a lack of evidence to support the claim that hedge funds, as a whole, caused the crash. Park et al. (1998) tested the hypothesis that hedge funds were responsible for the crash of Asian currencies in late 1997. The authors adapted the asset class factor model that is a common model used for investment analysis based on the previous study by Sharpe (1992) to analyse the hedge funds' returns. Sharpe found no empirical evidence to support the hypothesis that hedge funds were responsible for the crisis. Some recent studies also disapprove the claims of the existence of correlation and/or contagion between hedge funds and stock markets. For example, Sias et al. (2018) revealed that evidence of hedge fund contagion is quite scarce. The authors also suggest that, despite the potential of hedge funds to generate better returns compared with stock markets, the contagion effects need to be explored further. The authors also recognised that future events and the analysis of fund markets may reveal further evidence of hedge fund contagion. Kanuri (2020) studied the performance of hedge funds and compared the returns to the performance of the Japanese stock market returns from 2000–2018 and found evidence that hedge funds outperformed Japanese stocks and bonds market with much higher returns. Overall, their results indicated that hedge funds added much value for investors compared with the stocks and bonds market because of the lack of correlation in market movements.

Although there is a little evidence that the hedge funds contributed to the financial crisis, some researchers suggest that withdrawing assets from investment banks led to the collapse of these institutions and contributed to the 2008 Global Financial Crisis. Cao et al. (2018) investigated whether hedge fund leverage played a role in propagating the shocks to price efficiency following the failure of Lehman Brothers. They found evidence that mispricing of stocks following Lehman's failure was more severe for stocks held by hedge funds that use leverage.

A study by Adams et al. (2014) found evidence that hedge funds may be the most important transmitter of shocks during the 2008 Global Financial Crisis, more so than commercial or investment banks. Hedge funds are considered to be highly opaque and leveraged investment instruments. This property of hedge funds allows liquidation of assets at high prices, resulting in heavy losses to the asset classes involved. This led to further defaults through asset price adjustments Bernanke (2006).

To trace the spillover effects from hedge funds to assets classes, one would need detailed information on the different risks to which financial markets are exposed, their liabilities and their assets. Unfortunately, hedge funds are not required to supply this type of information. Concern over the systemic importance of hedge funds also emphasises tighter reporting obligations for large institutions in the Dodd-Frank Wall Street Reform and Consumer Protection Act of 2010 Lo (2008).

The results in this paper are important to policymakers in the Asia-Pacific region endeavouring to form macroeconomic policies. Similarly, the results can provide policymakers with additional tools to advance their efforts at maintaining financial market stability against potential information spillover impacts from stock markets and global financial markets. In parallel, the paper's results add to academic efforts to understand the extent of the linkages between financial markets. From investors' viewpoint, the paper gives a new perspective to accomplish investment diversification. Also, the varying levels of volatility spillover and correlation during stable and turbulent crisis periods enriches the literature on financial contagion theory in the context of the Asia-Pacific region.

**Author Contributions:** Conceptualization, S.F. and C.G.; methodology, S.F.; software, S.F.; validation, B.H.; formal analysis, S.F.; investigation, S.F. and B.H.; resources, S.F.; data curation, S.F.; writing—original draft preparation, S.F.; writing—review and editing, B.H.; visualization, S.F.; supervision, C.G. and B.H. All authors have read and agreed to the published version of the manuscript.

**Funding:** This research received no external funding.

**Conflicts of Interest:** The authors declare no conflict of interest.

## Appendix A

**Table A1.** Descriptive Statistics of 12 Asia-Pacific Countries' Stock Market Returns.

| Stock Market | Sample | Obs. | Mean | SD | Skewness | Kurtosis | JB |
|---|---|---|---|---|---|---|---|
| ASX | Total Sample | 1056 | 24.69579 | 12.93494 | −0.167332 | 1.797859 | 68.5143 |
| | 1998–2002 | 220 | 6.667713 | 1.100195 | −0.368906 | 5.134461 | 46.75268 |
| | 2003–2009 | 365 | 22.84487 | 11.06226 | 0.553851 | 2.726017 | 19.80235 |
| | 2010–2018 | 471 | 34.55091 | 5.301064 | 0.821111 | 3.122298 | 53.22 |
| NZX | Total Sample | 814 | 0.658846 | 0.279324 | −0.16943 | 2.010832 | 37.08043 |
| | 2003–2009 | 396 | 0.435861 | 0.196287 | 0.353459 | 2.043709 | 23.33474 |
| | 2010–2018 | 418 | 0.870096 | 0.153842 | 0.211961 | 1.787517 | 28.31383 |
| JPY | Total Sample | 1148 | 131.0165 | 32.10497 | 0.3189 | 2.71099 | 23.45341 |
| | 1997–2002 | 313 | 126.3965 | 32.82477 | 0.067423 | 1.988238 | 13.58741 |
| | 2003–2009 | 365 | 113.7442 | 22.82147 | −0.163269 | 2.082395 | 14.42702 |
| | 2010–2018 | 470 | 147.5069 | 29.67909 | 0.482603 | 2.283108 | 28.30881 |
| STI | Total Sample | 1010 | 1822.812 | 630.4495 | −0.293034 | 1.576597 | 99.7186 |
| | 1999–2002 | 174 | 1039.979 | 202.2187 | 0.284898 | 1.988593 | 9.770189 |
| | 2003–2009 | 365 | 1501.37 | 498.0532 | 0.366934 | 1.991985 | 23.64376 |
| | 2010–2018 | 471 | 2361.113 | 220.4931 | −0.40187 | 2.078595 | 29.33907 |
| NSE500 | Total Sample | 1149 | 65.29906 | 40.13882 | 0.166424 | 1.734851 | 81.93286 |
| | 1997–2002 | 313 | 18.42959 | 4.455568 | 1.528323 | 5.944602 | 234.9295 |
| | 2003–2009 | 365 | 57.68301 | 28.22967 | 0.531987 | 2.645008 | 19.13302 |
| | 2010–2018 | 471 | 102.3479 | 20.52733 | 0.462887 | 2.447648 | 22.80718 |
| KOSPI | Total Sample | 1147 | 1.270322 | 0.597327 | −0.12531 | 1.598694 | 96.84837 |
| | 1997–2002 | 313 | 0.565223 | 0.186539 | −0.115554 | 1.9739 | 14.4279 |
| | 2003–2009 | 365 | 1.171179 | 0.459903 | 0.336727 | 2.129651 | 18.41803 |
| | 2010–2018 | 469 | 1.818048 | 0.212553 | 0.416116 | 3.231304 | 14.58028 |
| SHCOMP | Total Sample | 1149 | 312.5864 | 159.2746 | 0.367888 | 2.711571 | 29.90068 |
| | 1997–2002 | 313 | 175.762 | 56.73049 | −1.15276 | 5.410229 | 145.0837 |
| | 2003–2009 | 365 | 289.5869 | 177.6396 | 0.903963 | 3.058546 | 49.76199 |
| | 2010–2018 | 471 | 421.3355 | 103.2406 | −0.580439 | 8.674866 | 658.4529 |
| HSI | Total Sample | 1149 | 2345.888 | 747.5231 | 0.09526 | 2.004987 | 49.13647 |
| | 1997–2002 | 313 | 1600.539 | 333.9865 | 0.183424 | 2.084872 | 12.67698 |
| | 2003–2009 | 365 | 2139.414 | 631.3236 | 0.581075 | 2.665013 | 22.24688 |
| | 2010–2018 | 471 | 3001.211 | 385.6299 | 0.77717 | 3.315208 | 49.36328 |
| SET | Total Sample | 1148 | 25.7302 | 15.17621 | 0.420828 | 1.800974 | 102.6527 |
| | 1997–2002 | 313 | 10.75358 | 5.637739 | 2.133786 | 7.130056 | 459.9734 |
| | 2003–2009 | 365 | 17.82547 | 4.364359 | −0.085507 | 2.835278 | 0.85743 |
| | 2010–2018 | 470 | 41.84277 | 8.29781 | −0.431012 | 2.822071 | 15.1721 |

**Table A1.** *Cont.*

| Stock Market | Sample | Obs. | Mean | SD | Skewness | Kurtosis | JB |
|---|---|---|---|---|---|---|---|
| JCI | Total Sample | 1148 | 0.235168 | 0.158254 | 0.11425 | 1.425431 | 121.0892 |
| | 1997–2002 | 313 | 0.081725 | 0.073627 | 2.040987 | 5.807762 | 320.1212 |
| | 2003–2009 | 365 | 0.152355 | 0.074311 | 0.363281 | 1.946834 | 24.89679 |
| | 2010–2018 | 470 | 0.401667 | 0.067272 | −2.820489 | 17.21729 | 4581.557 |
| FBMKLCI | Total Sample | 1149 | 339.0695 | 137.1726 | 0.104476 | 1.688539 | 84.43188 |
| | 1997–2002 | 313 | 209.5008 | 92.0831 | 1.897886 | 6.178057 | 319.6243 |
| | 2003–2009 | 365 | 278.3003 | 74.4692 | 0.606984 | 2.205425 | 32.01459 |
| | 2010–2018 | 471 | 472.2664 | 64.87676 | 0.096004 | 1.751033 | 31.33692 |
| TWSE | Total Sample | 1134 | 241.7837 | 61.74244 | −0.084668 | 2.323889 | 22.95407 |
| | 1997–2002 | 310 | 218.5893 | 62.65417 | 0.10035 | 2.012759 | 13.10945 |
| | 2003–2009 | 360 | 201.3879 | 43.12455 | 0.265095 | 2.526674 | 7.577086 |
| | 2010–2018 | 464 | 288.6214 | 37.93733 | 0.546348 | 2.503783 | 27.84416 |

ASX: Australian Stock Price, NZX: New Zealand Stock Price, JPY: Japanese Stock Price, STI: Singaporean Stock Price, NSE500: Indian Stock Price, KOSPI: South Korean Stock Price, SHCOMP: Shanghai China Stock Price, HSI: Hong Kong Stock Price, SET: Thai Stock Price, JCI: Indonesian Stock Price, FBMKLCI: Malaysian Stock Price, TWSE: Taiwanese Stock Price.

**Table A2.** Descriptive Statistics of Hedge Fund Returns.

| Hedge Funds | Obs. | Type of Test | | | | |
|---|---|---|---|---|---|---|
| | | Mean | SD | Skewness | Kurtosis | JB |
| AGF | 264 | 0.805265 | 3.600888 | −0.41628 | 5.674495 | 86.30688 |
| AACF | 265 | 1.148491 | 5.878306 | −0.190843 | 5.088338 | 49.763 |
| AAGF | 264 | 0.660227 | 7.111169 | 0.734775 | 5.776641 | 108.5625 |
| BDP | 256 | 0.506445 | 2.817598 | 0.215363 | 3.963545 | 11.88206 |
| BAF | 264 | 0.580379 | 5.370084 | −0.009871 | 4.683957 | 31.1971 |
| CFB-FE | 264 | 0.924583 | 3.961538 | 0.553681 | 6.501578 | 148.6302 |
| CFB-HK | 264 | 0.735985 | 3.774504 | −0.295734 | 6.680355 | 152.8434 |
| CFB-T | 264 | 1.171553 | 4.468795 | 0.255109 | 5.438744 | 68.28576 |
| HFNV | 252 | 1.042659 | 9.04983 | 0.107356 | 5.606503 | 71.81957 |
| HKP | 264 | 0.590871 | 7.82731 | 0.051869 | 4.303096 | 18.79704 |
| IIF | 264 | 1.380076 | 9.572629 | 0.068171 | 4.610727 | 28.74333 |
| IVI | 264 | 1.108598 | 6.734081 | −0.29915 | 5.056565 | 50.46164 |
| IGF | 260 | 1.190038 | 12.75944 | 0.166466 | 7.380011 | 209.0329 |
| JKAI | 264 | 0.593788 | 5.417733 | 0.324015 | 3.592816 | 8.485114 |
| LIM | 264 | 0.504356 | 1.629906 | 0.030212 | 10.74091 | 659.178 |
| MLM | 264 | 0.858371 | 4.696964 | 0.489854 | 3.314214 | 11.64415 |
| PPL | 264 | 0.819697 | 3.590945 | −0.102776 | 3.728723 | 6.306183 |
| PIF | 264 | 0.99072 | 3.269807 | 0.392461 | 4.00035 | 17.78484 |
| PCF | 264 | 0.775492 | 7.174266 | 0.273565 | 8.195353 | 300.2015 |
| ISF | 264 | 0.547121 | 6.736905 | −0.121409 | 4.888512 | 39.8798 |
| SJO | 264 | 0.915758 | 9.62004 | 0.935716 | 5.259409 | 94.67905 |
| SRG | 264 | 0.958598 | 6.025513 | 0.795778 | 5.514949 | 97.43818 |
| VPC-A | 264 | 1.172765 | 6.506845 | −0.680881 | 6.209624 | 133.7169 |
| VPC-B | 264 | 1.142273 | 6.539423 | −0.67732 | 6.106166 | 301.56 |
| VEI | 264 | 0.926818 | 6.639179 | 0.084498 | 5.514089 | 69.84124 |

AGF: Allard Growth Fund, AACF: Arisaig Asia Consumer Fund Ltd., AAGF: Atlantis Japan Growth Fund Ltd., BDP: Boronia Diversified Program, BAF: Bowen Asia Fund, CFB-FE: CFB Convertibles Fund PLC—Far East Sub Fund, CFB-HK: CFB Convertibles Fund PLC—Hong Kong Sub Fund, CFB-T: CFB Convertibles Fund PLC—Thailand Sub Fund, HFNV: Himalayan Fund NV, HKP: Hong Kong Partners LP, IIF: India Capital Fund Ltd.—A Share, IVI: India Value Investments Ltd.—GBP, IGF: Indonesian Growth Fund, JKAI: JK Asian Invest LP, LIM: LIM Asia Multi-Strategy Fund—Class A Series 1, MLM: MLM Macro—Peak Partners LP, PPL: Platinum Fund Ltd.—USD, PIF: Platinum International Fund—Class C, PCF: Polar Capital Funds plc—Asian Opportunities Fund Class USD, ISF: Schroder ISF Asian Opportunities—USD, SJO: Shiozumi Japan Opportunities Fund, SRG: SR Global Fund Class C) International, VPC-A: Value Partners Classic Fund—Class A USD, VPC-B: Value Partners Classic Fund—Class B USD, VEI: Vietnam Enterprise Investments Ltd.

**Table A3.** ADF Unit Root Test of Stock Markets Data.

| Type of Test | ADF | | PP | |
|---|---|---|---|---|
| **Stock Market** | **Levels** | **1st Diff.** | **Levels** | **1st Diff.** |
| ASX | −1.515675 | −9.279596 * | −1.475958 | −32.4942 * |
| NZX | −2.014037 | −28.36016 * | −2.052167 | −28.40898 * |
| JPY | −1.653262 | −37.10071 * | −1.751896 | −37.04085 * |
| STI | −1.413977 | −17.7805 * | −1.350301 | −30.11826 * |
| NSE500 | −0.963937 | −20.67192 * | −0.897202 | −31.31889 * |
| KOSPI | −1.276449 | −34.45373 * | −1.274097 | −34.44916 * |
| SHCOMP | −2.981296 | −5.010659 * | −2.102386 | −30.33593 * |
| HSI | −1.536054 | −22.81121 * | −1.610195 | −33.89964 * |
| SET | −0.611759 | −13.54181 * | −0.564731 | −33.42701 * |
| JCI | −0.844615 | −13.0041 * | −0.771935 | −36.28122 * |
| FBMKLCI | −1.616394 | −12.83962 * | −1.559871 | −33.97777 * |
| TWSE | −2.127654 | −22.31131 * | −2.102121 | −34.33078 * |

* 1% significance level, ASX: Australian Stock Price, NZX: New Zealand Stock Price, JPY: Japanese Stock Price, STI: Singaporean Stock Price, NSE500: Indian Stock Price, KOSPI: South Korean Stock Price, SHCOMP: Shanghai China Stock Price, HSI: Hong Kong Stock Price, SET: Thai Stock Price, JCI: Indonesian Stock Price, FBMKLCI: Malaysian Stock Price, TWSE: Taiwanese Stock Price.

**Table A4.** ADF Unit Root Test Hedge Funds.

| Type of Test | | ADF | |
|---|---|---|---|
| **Hedge Fund** | **Variable** | **Level** | **1st Diff.** |
| Allard Growth Fund | AGF | −6.820719 | −12.36761 |
| Arisaig Asia Consumer Fund Ltd. | AACF | −6.145342 | −10.69169 |
| Atlantis Japan Growth Fund Ltd. | AAGF | −5.868159 | −10.0679 |
| Boronia Diversified Program | BDP | −6.981794 | −11.30284 |
| Bowen Asia Fund | BAF | −6.826534 | −10.24941 |
| CFB Convertibles Fund PLC—Far East Sub Fund | CFB-FE | −5.464945 | −9.970182 |
| CFB Convertibles Fund PLC—Hong Kong Sub Fund | CFB-HK | −7.226543 | −10.70699 |
| CFB Convertibles Fund PLC—Thailand Sub Fund | CFB-T | −6.316322 | −9.140703 |
| Himalayan Fund NV | HFNV | −6.84573 | −11.68212 |
| Hong Kong Partners LP | HKP | −6.609208 | −11.1289 |
| India Capital Fund Ltd.—A Share | IIF | −6.581257 | −12.35492 |
| India Value Investments Ltd.—GBP | IVI | −5.64695 | −13.23924 |
| Indonesian Growth Fund | IGF | −6.00082 | −11.25764 |
| JK Asian Invest LP | JKAI | −5.977589 | −10.8305 |
| LIM Asia Multi-Strategy Fund—Class A Series 1 | LIM | −7.133593 | −10.8006 |
| MLM Macro—Peak Partners LP | MLM | −7.672718 | −12.47703 |
| Platinum Fund Ltd.—USD | PPL | −6.604396 | −13.36312 |
| Platinum International Fund—Class C | PIF | −5.955776 | −9.81512 |
| Polar Capital Funds plc—Asian Opportunities Fund Class USD | PCF | −7.010237 | −10.70807 |
| Schroder ISF Asian Opportunities—USD A Dis | ISF | −6.48012 | −10.63692 |
| Shiozumi Japan Opportunities Fund | SJO | −6.397172 | −12.12368 |
| SR Global Fund Class C) International | SRG | −7.443766 | −12.2378 |
| Value Partners Classic Fund—Class A USD | VPC-A | −7.931334 | −10.79921 |
| Value Partners Classic Fund—Class B USD | VPC-B | −7.920528 | −10.83225 |
| Vietnam Enterprise Investments Ltd. | VEI | −7.879385 | −10.55903 |

**Table A4.** *Cont.*

| Type of Test | | PP | |
|---|---|---|---|
| Hedge Fund | Variable | Level | 1st Diff. |
| Allard Growth Fund | AGF | −6.20142 | −13.53153 |
| Arisaig Asia Consumer Fund Ltd. | AACF | −5.312271 | −15.93447 |
| Atlantis Japan Growth Fund Ltd. | AAGF | −5.009445 | −13.99729 |
| Boronia Diversified Program | BDP | −4.404156 | −15.70672 |
| Bowen Asia Fund | BAF | −4.98403 | −17.64692 |
| CFB Convertibles Fund PLC—Far East Sub Fund | CFB-FE | −4.710158 | −16.99101 |
| CFB Convertibles Fund PLC—Hong Kong Sub Fund | CFB-HK | −4.830855 | −18.77841 |
| CFB Convertibles Fund PLC—Thailand Sub Fund | CFB-T | −5.32639 | −12.87429 |
| Himalayan Fund NV | HFNV | −4.957175 | −17.93565 |
| Hong Kong Partners LP | HKP | −5.17482 | −25.41986 |
| India Capital Fund Ltd.—A Share | IIF | −5.158558 | −26.4707 |
| India Value Investments Ltd.—GBP | IVI | −5.031586 | −29.81622 |
| Indonesian Growth Fund | IGF | −5.647025 | −10.98228 |
| JK Asian Invest LP | JKAI | −5.014096 | −18.33084 |
| LIM Asia Multi-Strategy Fund—Class A Series 1 | LIM | −6.01404 | −11.50953 |
| MLM Macro—Peak Partners LP | MLM | −4.780957 | −16.21311 |
| Platinum Fund Ltd.—USD | PPL | −5.151409 | −18.95169 |
| Platinum International Fund—Class C | PIF | −5.133875 | −16.82738 |
| Polar Capital Funds plc—Asian Opportunities Fund Class USD | PCF | −5.001961 | −25.78604 |
| Schroder ISF Asian Opportunities—USD A Dis | ISF | −4.94628 | −19.52952 |
| Shiozumi Japan Opportunities Fund | SJO | −5.038999 | −17.60632 |
| SR Global Fund Class C) International | SRG | −5.237148 | −23.91525 |
| Value Partners Classic Fund—Class A USD | VPC-A | −4.773421 | −16.72473 |
| Value Partners Classic Fund—Class B USD | VPC-B | −4.778884 | −16.69193 |
| Vietnam Enterprise Investments Ltd. | VEI | −4.75629 | −19.39658 |

* 1% significance level.

**Table A5.** Diagnostics for the EGARCH Residuals—Stock Returns.

| Country | ASX | NZX | JPY | STI | NSE500 | KOSPI |
|---|---|---|---|---|---|---|
| JB | 104 | 63 | 5012 | 75 | 15 | 211 |
| LB 20 | 27.789 | 26.688 | 24.015 | 65.244 | 54.077 | 15.978 |
| | −0.127 | −0.151 | −0.211 | 0.000 | 0.000 | −0.644 |
| LB$^2$ 20 | 20.262 | 21.717 | 15.566 | 4.1955 | 3.4736 | 13.796 |
| | −0.452 | −0.414 | −0.849 | −1 | −1 | −0.896 |
| Country | SHCOMP | HSI | SET | JCI | FBMKLCI | TWSE |
| JB | 216 | 82 | 139 | 1068 | 275 | 94 |
| LB 20 | 42.066 | 43.904 | 45.743 | 47.581 | 49.419 | 51.258 |
| | −0.3809 | −0.4358 | −0.4907 | −0.5456 | −0.6005 | −0.6554 |
| LB$^2$ 20 | 3.3253 | 0.513 | −2.2993 | −5.1116 | −7.9239 | −10.736 |
| | −1.1814 | −1.2994 | −1.4173 | −1.5353 | −1.6533 | −1.7713 |

**Table A6.** Diagnostics for the EGARCH Residuals—Hedge Funds Returns.

| Hedge Fund | AGF | AACF | AAGF | BDP | BAF | CFB-FE | CFB-HK | CFB-T | HFNV | HKP | IIF | IVI | IGF | JKAI | LIM | MLM | PPL | PIF | PCF | ISF | SJO | SRG | VPC-A | VPC-B | VEI |
|---|---|---|---|---|---|---|---|---|---|---|---|---|---|---|---|---|---|---|---|---|---|---|---|---|---|
| JB | | | | | | | | | | | | | | | | | | | | | | | | | |
| LB 20 | 27.8<br>−0.1 | 26.7<br>−0.2 | 24.0<br>−0.2 | 65.2<br>0.0 | 54.1<br>0.0 | 16.0<br>−0.6 | 61.2<br>0.0 | 16.8<br>−0.6 | 27.4<br>−0.2 | 16.1<br>−0.5 | 24.8<br>−0.2 | 27.5<br>−0.1 | 29.7<br>−0.5 | 27.0<br>−0.1 | 19.1<br>−0.1 | 30.4<br>−0.5 | 68.2<br>−0.1 | 64.2<br>−0.1 | 19.8<br>−0.5 | 57.1<br>0.0 | 19.0<br>−0.1 | 30.5<br>−0.5 | 30.0<br>0.0 | 32.7<br>0.0 | 27.8<br>−0.4 |
| LB² 20 | 20.3<br>−0.5 | 21.7<br>−0.4 | 15.6<br>−0.8 | 4.2<br>−1.0 | 3.5<br>−1.0 | 13.8<br>−0.9 | 4.5<br>−1.0 | 6.0<br>−0.9 | 7.3<br>−0.9 | 7.7<br>−1.0 | 8.0<br>−1.0 | 7.5<br>−0.9 | 4.2<br>−0.4 | 5.7<br>−0.8 | 6.9<br>−1.0 | 7.3<br>−1.0 | 7.6<br>−0.9 | 3.4<br>−1.0 | 13.8<br>−0.9 | 4.5<br>−0.9 | 6.0<br>−1.0 | 7.2<br>−1.0 | 7.6<br>−0.9 | 7.9<br>−0.4 | 7.5<br>−0.8 |
| ARCH-LM | 1.0<br>−0.5 | 0.9<br>−0.4 | 0.8<br>−0.9 | 0.1<br>−1.0 | 0.1<br>−1.0 | 0.6<br>−0.8 | 0.3<br>−1.0 | 0.4<br>−1.0 | 0.5<br>−1.0 | 0.6<br>−0.9 | 0.5<br>−0.9 | 0.4<br>−1.0 | 0.3<br>−0.8 | 0.4<br>−1.0 | 0.5<br>−1.0 | 0.4<br>−1.0 | 0.3<br>−1.0 | 1.0<br>−1.0 | 0.9<br>−1.0 | 0.8<br>−0.8 | 0.1<br>−1.0 | 0.1<br>−0.3 | 0.6<br>−1.0 | 0.3<br>−1.0 | 0.4<br>−1.0 |

## Notes

[1]   The descriptive statistics for the stock market indices are provided in Appendix A.

[2]   The descriptive statistics for hedge fund returns' results are provided in Appendix A.

[3]   The ADF Unit Root Test (Hedge Funds) results are provided in Appendix A.

[4]   The testing results are not presented but are available upon request.

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
