# Peer review of "Volatility Spillovers between Stock Market and Hedge Funds: Evidence from Asia Pacific Region"

_jrfm, doi:10.3390/jrfm15090409_

Round 1
Reviewer 1 Report
Review of the article “Volatility Spillovers Between Stock Market and Hedge Funds: Evidence from Asia Pacific Region”
This paper investigates the relationships and volatility linkages between stock and hedge fund markets. Overall, this study makes contributions to the current literature. I am overall positive regarding the work, which is why I would be open to reviewing a revised manuscript, however, in my opinion, some revisions are necessary. Below presented some detailed comments. Hopefully they can assist the authors to improve the quality of the paper.
1. The Introduction should further motivate the study. Why this study is necessary? Does the study address any policy level problem? Is the study expected to provide any solution to that problem? How the choice of sample is complementing that problem? Are the results and policies generalizable? In the introduction the authors need to discuss latest empirical work to improve the motivation of the paper and broaden the view of readers and display the contribution of this paper.
2. In the Data and Methodology section the authors refer to VAR model. Results of some tests however have not been presented, e.g. Johansen test. Moreover, the authors claim that “the model is free from serial correlation” (p. 4), however, they do not present the results of the tests that support this conclusion.
3. In the same section the authors do not present and comment on the results of tests referring to the normality and stability of residuals in the VAR system. As regards the normality of the residuals eq. 1.3 and 1.4 are included in the study but readers of the article do not know if this is assumption or the observation based on the data analyzed.
4. Significant part of the study is in the form of tables which present volatility spillovers. I recommend to place these tables in the appendix and in the main part of the article put only the graphs showing effects of the spillovers.
Author Response
Dear Reviewer,
Please find our responses to your comments and suggestions.
Thank you

Reviewer 2 Report
Authors forgot to discus spillovers in crypto-currency and forex markets (in Introduction).
They claim that "Our study focusses on investigating the levels and magnitude of connections and provides a comprehensive, detailed summary...", however, to show magnitude of connections graph or network tools should be used instead of time series.
Some important factors, e.g. Volatility Persistence, are analyzed however, definitions and what they show are not provided in methodology.
Speaking about time series, authors forgot to check necessary conditions for EGARCH (provided in eq. 1.3 and 1.4).
Authors did not provided correlations between returns analyzed. If returns are highly correlated then it may turn out that all analysis performed is useless. Moreover, if correlations are very different in different periods then it can turn out that different market regimes are observed and data set should be splitted.
From tables A1 ad A2 it is clear that returns are not symmetrically distributed, therefore I really doubt that symmetrical spillover has logical sense. Moreover, this suggest that conclusions can be incorrect.
Tables 1-13 are terrible. I insist to transpose these tables and leave only important information (please remove numbers in brackets). By the way, why in some cases numbers in brackets are negative (e.g. Thailand, -0.291,-0.290)?
Author Response
Dear Reviewer,
Please find our responses to your comments and suggestions.
Thanks you

Round 2
Reviewer 2 Report
My previous comment 7 is still unsolved.
By saying please "please remove numbers in brackets" I meant remove brackets and numbers as they do not provide anything important.
By the way, why in some cases numbers in brackets are negative (e.g., Thailand, -0.291, -0.290)? Significance cannot be negative!!!!
Author Response
Dear Reviewer,
Thank you for pointing out the typos in the tables. We have now corrected them.
Sincerely,
The authors
